# CODESENSE: A REAL-WORLD BENCHMARK AND DATASET FOR CODE SEMANTIC REASONING

**Monoshi Kumar Roy**[1]**, Simin Chen**[2]**, Benjamin Steenhoek**[3]**, Jinjun Peng**[2]**,
Gail Kaiser**[2]**, Baishakhi Ray**[2]**, Wei Le**[1]
[1]Iowa State University, {monoshi, weile}@iastate.edu
[2]Columbia University, {sc5687, jinjun, kaiser, rayb}@cs.columbia.edu
[3]Microsoft Data & AI, bensteenhoek@microsoft.com

## ABSTRACT

Understanding and reasoning about code semantics is essential for enhancing code LLMs' abilities to solve real-world software engineering (SE) tasks. Although several code reasoning benchmarks exist, most rely on synthetic datasets or educational coding problems and focus on coarse-grained reasoning tasks such as input/output prediction, limiting their effectiveness in evaluating LLMs in practical SE contexts. To bridge this gap, we propose CodeSense, the first benchmark that makes available a spectrum of fine-grained code reasoning tasks concerned with the software engineering of real-world code. We collected Python, C and Java software projects from real-world repositories. We executed tests from these repositories, collected their execution traces, and constructed a ground truth dataset for fine-grained semantic reasoning tasks. We then performed comprehensive evaluations on state-of-the-art LLMs. Our results show a clear performance gap for the models to handle fine-grained reasoning tasks. Although prompting techniques such as chain-of-thought and in-context learning helped, the lack of code semantics in LLMs fundamentally limits models' capabilities of code reasoning. Besides dataset, benchmark and evaluation, our work produced an execution tracing framework and tool set that make it easy to collect ground truth for fine-grained SE reasoning tasks, offering a strong basis for future benchmark construction and model post training. Our code and data are located at https://codesense-bench.github.io/.

## 1 INTRODUCTION

Semantic code reasoning—the capacity to understand and predict the behaviour of software—is a core requirement underpinning a wide range of complex software engineering (SE) tasks, including test input generation, vulnerability detection, fault localization, bug repair, refactoring, and functional verification. Unlike syntactic pattern matching, which may rely on token-level similarity or statistical regularities, semantic reasoning ("codesense") entails a deep, execution-oriented understanding of how software operates. Although code semantics can be expressed in many ways, in practice, developers engage in semantic reasoning through tasks like predicting a function's input-output behavior, tracing variable values, analyzing control flow paths, identifying loop invariants, etc. This form of reasoning aligns with formal definitions from programming language theory—particularly operational semantics, which models step-by-step execution, and axiomatic semantics, which uses logical assertions to describe program properties. Such reasoning tasks also reflect the real-world demands placed on developers and provide a natural grounding for their day-to-day work.

Recent years have witnessed the emergence of numerous benchmarks for evaluating coding-related tasks. However, the majority of these efforts have focused on code generation using synthetic or narrowly scoped data—for example, HumanEval+ (Liu et al., 2023), LiveCodeBenchmark (Jain et al., 2024), Bigcodebench (Zhuo et al., 2024), and CodeBenchGen (Xie et al., 2024)—often extracted from isolated competitive programming problems. Consequently, they fail to capture the complexity and structure of real-world software development. Other benchmarks that incorporate real-world code, such as SWE-Bench (Jimenez et al., 2024), SWE-PolyBench (Rashid et al., 2025), and KGym (Mathai et al., 2024), tend to evaluate only task-specific performance (e.g., patch generation for GitHub issues), making it difficult to assess whether models exhibit generalizable semantic understanding. Finally,

```
1    void foo(int input){
2        int n = input * 23;
3        if (3465>=n>=2287){
4            //dangerous code need to be tested
5        }
6    }
```

```
1    void bar (int nbits) {
2        FFTContext *s = malloc(sizeof (* s));
3        if (s && nbits==100))
4            free(& s);
5        else ... return s;
6    }
```

(A) Test input generation and program execution reasoning: to generate a test input that can lead to the execution of dangerous code at line 4, the model needs to find an input that can satisfy the branch condition $3465 \geq n \geq 2287$ at line 3. Understanding the semantics of operators '*' at line 2 and '$\geq$' at line 3 is needed to effectively generate an input that reaches the dangerous code, e.g., input = 120 . The branches and arithmetic operations can be quite diverse in different programs, and it is hard to generalize patterns regarding which code text should use what kind of test input to execute.

(B) Vulnerability detection, fault localization and program repair: this code has a memory leak vulnerability when the `if` condition at line 3 is `false`. To detect this vulnerability, the model should know that calls to `malloc` and `free` are related to the vulnerability (semantics of the API calls), and should be paired along the program paths along both branches starting at line 3 (semantics of the branch statements and control flow). Similarly, `malloc` and `free` can be located in various code contexts in different programs, and thus it is hard to generalize the patterns only from the code text, without semantic code reasoning.

FIGURE 1. Fine-grained code semantics are the keys for solving many SE tasks

reasoning-focused benchmarks, such as CruxEval (Gu et al., 2024), primarily target function-level input/output prediction over short and synthetic code fragments involving random string operations. Such settings neglect the fine-grained semantic reasoning about internal program behavior and properties, data dependencies, and control structures required to solve a variety of SE tasks for complex real-world software systems.

To this end, we propose CodeSense, a benchmark for fine-grained code semantic reasoning, constructed from real-world GitHub projects in Python, C, and Java (see Table 1). CodeSense introduces a spectrum of reasoning tasks at statement, code-block, and function levels, targeting essential semantic properties frequently needed across SE activities. For example, predicting loop iteration counts is critical for input/output prediction, performance analysis, and detecting infinite loops (e.g., denial-of-service vulnerabilities). Branch condition prediction and reasoning about pointers in C code are important for test input generation and memory safety assurance. As illustrated in Figures 1a and 1b, fine-grained semantic reasoning about arithmetic operations, control flow, and API semantics is foundational for a variety of SE applications. Prior work (Ding et al., 2023a; 2024) has shown that incorporating semantic signals during training improves model performance on code generation, branch prediction, code clone detection, program repair and vulnerability detection tasks, motivating our design of CodeSense to comprehensively evaluate models' capabilities for semantic reasoning.

TABLE 1. Optimal design space of code reasoning benchmarks (○ denotes not support, ◐ denotes partial support, and ● denotes fully support)

| Benchmark | Real-World Projects | Multi-lingual | Function I/O | Fine-Grained Reasoning | Exec Steps | API Understanding | Multi-File Context |
|---|---|---|---|---|---|---|---|
| CruxEval (Gu et al., 2024) | ○ | ○ | ● | ○ | ○ | ○ | ○ |
| CruxEval-X (Xu et al., 2025) | ○ | ● | ● | ○ | ○ | ○ | ○ |
| REval (Chen et al., 2024) | ○ | ○ | ● | ◐ | ◐ | ○ | ○ |
| CodeMind (Liu et al., 2024) | ● | ● | ● | ○ | ○ | ● | ● |
| CoRe (Xie et al., 2025) | ○ | ● | ○ | ● | ○ | ○ | ○ |
| CodeSense | ● | ● | ● | ● | ● | ● | ● |

To construct CodeSense, we collected 544 Python, 100 C, and 100 Java projects. To enable supervised evaluation, we develop a framework that automatically extracts ground-truth annotations for these tasks using static and dynamic program analysis. We build, execute and log run-time execution values and traces of the real-world projects. Our dataset includes a total of 2125 Python, 876 C, and 875 Java unique functions, based on which, we curated 4483 samples with their ground truth in our benchmark. Please see more details in Section 2.

Using our benchmark, we evaluated 14 state-of-the-art (SOTA) LLMs and investigated six research questions regarding the models' code semantics reasoning capabilities. Previous work has shown that models did not perform well on code reasoning tasks such as input/output prediction (Gu et al., 2024) and vulnerability detection (Steenhoek et al., 2025); to understand why models fail and identify places for improvement, we investigate: **RQ1:** Does increasing code size make semantic reasoning more difficult? **RQ2:** Which types of program statements are easier or harder for models to reason about? **RQ3:** How do models perform on code properties critical for SE tasks, such as predicting pointer aliasing, loop iteration counts, and branch conditions? **RQ4:** How effective are different prompting strategies in improving semantic reasoning? **RQ5:** Can models reason approximately when exact values or semantics are hard to infer? **RQ6:** Do models perform better on some programming languages than others?

Our results reveal that current LLMs, including SOTA models like Claude 3.5 (Anthropic, 2024), GPT-4o-mini (OpenAI, 2024), and Gemini 1.5 (Gemini Team et al., 2024) struggle with fine-grained code semantics. They often fail to reason about even single statements from real-world code—particularly arithmetic expressions and API calls—and perform poorly on tasks involving loop values and iteration counts. Basic chain-of-thought prompting offers limited benefit, and few-shot prompting yields only modest improvements. In-context learning is most effective when prompts define new concepts or include highly relevant examples. Interestingly, models can correlate natural language code semantics questions with certain code patterns. For instance, when the code contains assignments like `p = q`, models correctly respond to the prompt "do `p` and `q` at <line> alias the same memory address" even in zero-shot settings. Similarly, models reliably infer loop bounds in explicit cases such as `for i in range(100):`. Among the 14 models evaluated, Claude 3.5 consistently achieved the best performance. We also observe that Java and Python code are generally easier for models to reason about than C, and that input prediction (i.e., reverse semantic inference) remains among the most challenging tasks.

**Contributions.** This work introduces CodeSense, a realistic and comprehensive benchmark for evaluating LLMs' fine-grained code semantics reasoning in practical software engineering contexts. We advance the state-of-the-art code reasoning benchmarks by:

1. Defining a diverse set of fine-grained semantic reasoning tasks grounded in real-world software engineering needs,
2. Developing a scalable open-source framework and toolchain to automatically generate execution traces and semantic annotations, enabling continuous benchmark expansion while mitigating data leakage,
3. Constructing a benchmark dataset using real-world projects in Python, C, and Java,
4. Empirically analyzing six research questions across 14 state-of-the-art LLMs to assess their strengths and limitations in semantic reasoning, and
5. Launching a public leaderboard to support reproducibility and accelerate progress on semantic reasoning for code: `https://codesense-bench.github.io/leaderboard.html`

## 2 Benchmark Construction

### 2.1 Defining a Spectrum of Code Reasoning Tasks

To design tasks for evaluating LLMs' capabilities of code semantic reasoning, we first considered the definition of code semantics. In programming languages and software engineering, code semantics —"what is the meaning of this code" — are defined as what is the output value given the input of a code snippet. Such fine-grained reasoning tasks are directly related to end-tasks in software engineering. For example, previous work (Ding et al., 2023a) shows that when fine-tuned with statement-level values, the performance of the models improved for vulnerability detection, branch prediction and code clone detection. Prior study (Steenhoek et al., 2025) reported that although recent LLMs improved math reasoning and natural language reasoning significantly, they are still insufficient for handling end-tasks related to code reasoning. To help locate the weakness of models' code reasoning at a fine-granularity and help models to improve a variety of SE applications that are linked to the fine-grained reasoning steps, we designed the following code reasoning tasks:

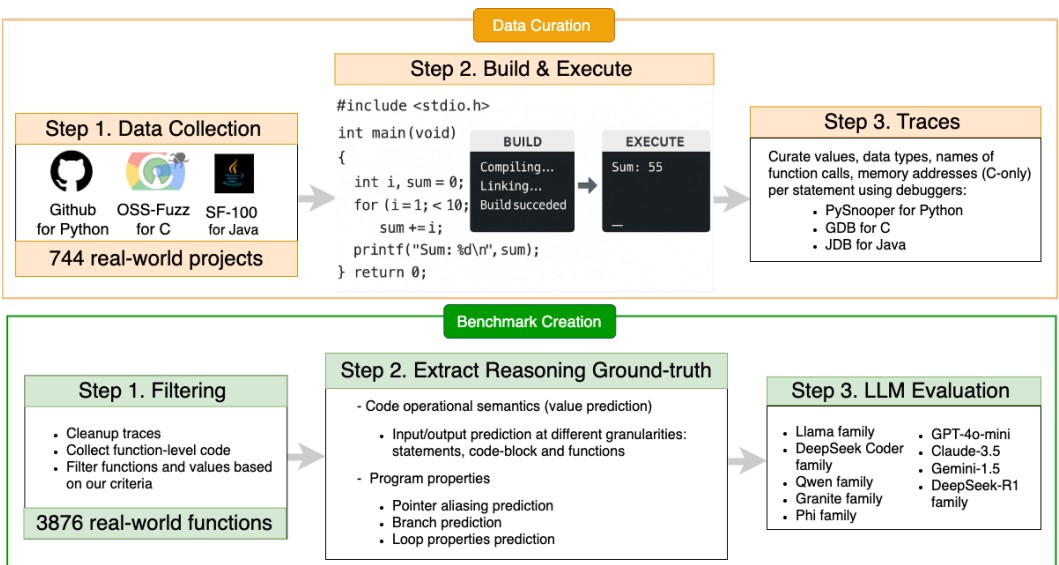

FIGURE 2. CodeSense: Data curation and benchmark creation overview.

**Task 1: Block-level code semantics (RQ1)**: To investigate whether a model understand a chunk of code, we give a block of statements. We give input and ask the models to predict the execution output; also we give output, we ask the models to predict the input. Input/output prediction of a function is a special case of probing block-level code semantics. In our evaluation, we sampled a block of statements from the entry of the functions and increased the sizes of blocks, including the entire function.

**Task 2: Statement-level code semantics (RQ2)**: We classified program statements based on programming language semantics and evaluated models on five common statement types, including *arithmetic*, *boolean expression*, *API/Function call*, *variable assignment* and *constant assignment*. Knowing code semantics at the statement level means that given an input of the statement, the models are able to produce a correct output. In our evaluation, we studied output predictions in more depth. We randomly sampled a statement from the program and asked the models to predict its output given input.

**Task 3: Code properties within a function (RQ3)**: Code property (property regarding a particular code construct) is another aspect of code semantics. We focused on three important properties in this benchmark. Loops are related to code optimization and detecting bugs. Reasoning about pointers in C code are very important for assuring memory safety and detecting and repairing vulnerabilities. Knowing how to predict branch outcomes can help generate test inputs and parallelize code.

**Task 3-1: Loop property.** Given an input of a function, we asked models to predict the number of loop iterations, the values in the loop and the values after executing the loop. In our evaluation, we randomly sampled a loop in a function and randomly sampled variables in and after the loop.

**Task 3-2: Pointer property.** Here, we give the models a function and its input, and we ask models to predict whether the two pointers are aliased (pointing to the same memory location) at a given program point.

**Task 3-3: Branch property.** Given a function and its input as well as the location of a conditional branch in the function, we ask the model to predict what is the outcome of the branch. In our evaluation, we randomly selected a conditional branch in the function for prediction.

**Task 4: Approximation of code semantics (RQ5)**: Reasoning about concrete values for the above tasks is very challenging. Sometimes, to solve an SE task, we may only need an approximate value of code semantics. For example, in Fig. 1a, the models do not have to generate a concrete number like `input=120`; it is sufficient for models to tell us that an integer input between 100-150 can trigger

the dangerous code. We designed a set of *abstract values* for different data types, following prior literature (Ding et al., 2023a) and evaluate if the models can predict abstract values correctly.

The above tasks are also used for studying **RQ4** regarding prompting techniques and **RQ6** comparing different programming languages. We have included the prompts for all the above tasks in Appendix/data package.

## 2.2 Collecting and Tracing Real-world Multi-Lingual Software Projects

Constructing ground truth for the set of code-semantics tasks is a great challenge. We collected a total of 744 real-world software projects of Python, C and Java from GitHub. We developed a framework and tool chain to build the projects, run tests and collect the execution traces which contain values, data types, names of function calls and memory addresses (for C code) at each statement. We developed analysis tools to extract ground truth for the benchmark tasks from those fine-grained code semantics data. Please check our Appendix A.4 for our language selection rationale and how our framework can be easily extended to other languages.

**Python.** We collected 1489 GitHub repositories from the PyPIbugs dataset (Allamanis et al., 2021). We removed projects that don't contain test cases or have not been updated in the last four years, and obtained 544 projects. We first installed dependencies for each project and used `pytest` (Krekel et al., 2004) to run tests, and `Pysnooper` (Rachum et al., 2019) for tracing.

**C.** We used 100 projects curated in OSS-Fuzz (Arya et al., 2023). We built and fuzzed the real-world projects using the OSS-Fuzz infrastructure in the docker environment with project-wise fuzzing harnesses. We developed a tracing framework built on the GNU debugger (GDB) (Free Software Foundation).

**Java.** We collected 100 projects from the SF110 dataset (Fraser & Arcuri, 2012). We used Evo-Suite (Fraser & Arcuri, 2011) to generate and run test cases and developed our tracing tool on top of Java Debugger (Oracle Corporation, 2025) to record the execution details of the projects.

## 2.3 Data Filtering

We collected whole program traces, from which we curated unique functions based on their entry and exit points in the execution trace logs. We excluded functions that only contain comments, too lengthy to fit into the models' context, and the functions which don't have meaningful functionality in their body. For example, some functions only contain one statement like "return 0", or "printf("...")" or some functions are just a wrapper for another function which we have tested, such as "void myfunc(){ func();}". We obtained a total of 2125 Python, 876 C and 875 Java unique functions, with the sizes ranging from 3 to 516 lines of code. From these unique functions, we curated our task-specific datasets.

In real-world code, we face many complexities, e.g., the input of a function and the values of a variable can be complex types. As the first step of probing models to reason about fine-grained code semantics for real-world code, we focus on ground truth values of primitive data types in all tasks, including `int`, `float`, `str`, `bool`, `list`, `pointer`, `double`, `dictionary`, `tuple`, etc. In the evaluation, we show that even for values of primitive types, the models find it challenging to predict them. We collected a total of 4483 samples from Python, C and Java, and constructed the ground truth for the above tasks and used for evaluation. See Table 2.

Table 2. Number of Samples for Tasks

| Task | Python | C | Java | Total Samples |
|---|---|---|---|---|
| Task 1: Block | 1860 | 731 | – | 2591 |
| Task 1: Function | 308 | 94 | 74 | 476 |
| Task 2/Task 4: Statement | 545 | 485 | – | 1030 |
| Task 3-1/Task 4: Loop | 105 | – | – | 105 |
| Task 3-2: Pointer | – | 49 | – | 49 |
| Task 3-3: Branch | 232 | – | – | 232 |
| Total Samples | 3050 | 1359 | 74 | 4483 |

## 3 EVALUATION

We evaluate 14 SOTA LLMs, 8 reasoning models and 6 non-reasoning models (Table 3 for full names and short IDs used in figures), including open-source models (Llama, phi), close-source/API models (GPT-4.0 Mini, Claude 3.5 and Gemini 1.5) and distilled models (DeepSeek R1 series), with the model parameter sizes ranging from 7 to 14 billions. We utilized vLLM(v0.3.3) as our inference engine to run the models.

TABLE 3. Model Names and Their IDs in the figures

| Full Model Name | Model ID in the figures |
|---|---|
| openai/gpt-4o-mini | GPT-4o (Reasoning) |
| anthropic.claude-3-5-sonnet-20241022-v2:0 | CL-3.5 (Reasoning) |
| gemini-1.5-flash-002 | Gem-1.5 (Reasoning) |
| meta-llama/Llama-3.1-8B-Instruct | L-3.1 |
| Qwen/Qwen2.5-14B-Instruct-1M | Q-2.5 |
| Qwen/Qwen2.5-Coder-7B-Instruct | Qwen2.5-C |
| deepseek-ai/DeepSeek-Coder-V2-Lite-Instruct | DS-C |
| microsoft/Phi-4-mini-instruct | Phi-4 |
| microsoft/Phi-3.5-mini-instruct | Phi-3.5 |
| ibm-granite/granite-3.2-8b-instruct | Gr-3.2 (Reasoning) |
| deepseek-ai/DeepSeek-R1-Distill-Qwen-7B | DSR1-Q-7B (Reasoning) |
| deepseek-ai/DeepSeek-R1-Distill-Llama-8B | DSR1-L (Reasoning) |
| deepseek-ai/DeepSeek-R1-Distill-Qwen-14B | DSR1-Q-14B (Reasoning) |
| ibm-granite/granite-3.2-8b-instruct-preview | Granite-3.2 Pr (Reasoning) |

We designed five different natural language prompt templates (see Appendix/data package), and ran them on a sampled dataset for each model. We observed that prompt templates are model-sensitive, but not task-sensitive. So we select a template for each model for all the tasks. We prompted the models to give a response inside specific tags (<ans> </ans>) and considered the response inside that tag to compare with the ground truth, as done in (Gu et al., 2024). For our evaluation metrics, we used accuracy (exact matching of the generated outputs of the models and the ground truth label).

In the following, we presented a selection of interesting results. For clarity, we present results from one representative LLM per model family to ensure model diversity. Please refer to the Appendix for the complete set of experimental results.

### 3.1 RESULTS FOR RQ1: BLOCK-LEVEL CODE SEMANTICS

Fig. 3 shows input/output prediction results for code blocks and functions (a special case of code block) across varying sizes. In Fig. 3a shows the results for three block sizes - blocks containing one, two, and three statements, respectively.

Overall, we observe that model accuracy is low even for small code blocks. For example, in C dataset, models such as Claude 3.5 and GPT-4o-mini achieve under 30% accuracy on single-statement blocks. Python yields slightly better results, though no model exceeds 50% accuracy. Performance further declines as block size increases from 1 to 3 statements, with open-source models performing significantly worse. This degradation stems from two primary challenges: models often fail to reason about individual statements, and they struggle to track variable state across statements. Notably, even Claude 3.5 achieves only 20% accuracy on 3-statement Python blocks, and less than 10% on C. However, in some cases, smaller blocks can be harder because they contain API calls.

A similar trend is observed in Fig. 3b (right: Output Prediction), where models perform better on smaller functions than larger ones for output prediction. However, performance on input prediction remains consistently poor (left figure) across all function sizes. This highlights a broader limitation: LLMs are even less capable of understanding the "reverse" of operational semantics, i.e., inferring inputs from outputs. Even the best-performing model, Claude 3.5, achieves only around 12% accuracy in input prediction for small functions.

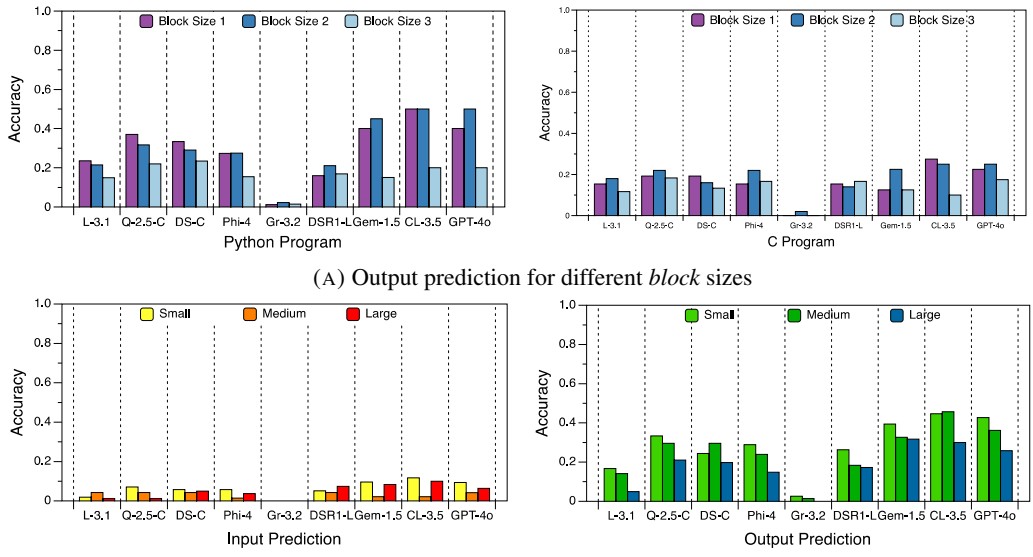

(A) Output prediction for different *block* sizes

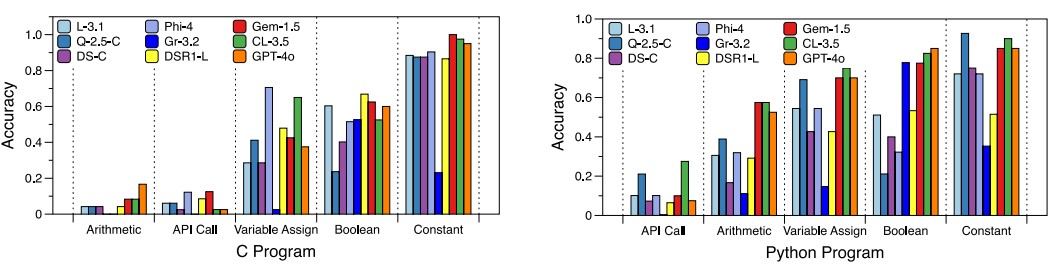

(B) In/Out prediction for different *function* sizes

FIGURE 3. **RQ1:** Does increasing the size of code increase the difficulty of code reasoning?

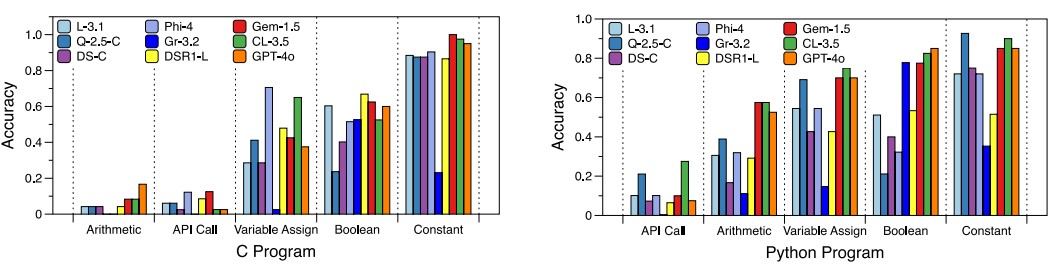

FIGURE 4. **RQ2:** What types of program statements can the model understand well?

## 3.2 RESULTS FOR RQ2: STATEMENT-LEVEL CODE SEMANTICS

As shown in RQ1, models struggle with value prediction even for single statements (e.g., block size 1). In RQ2, we further analyze model performance by categorizing results based on statement types. Fig. 4 presents these results, with the left plot showing C and the right showing Python. Each plot groups model performance by statement type to highlight specific areas of strength and weakness.

We observe that arithmetic and API/calls are the most difficult statement types, even for the best reasoning models like GPT4.0-mini and Claude 3. For APIs, we sampled frequently used third-party libraries, like `os`, `sys`, and `math` installed by `pip`, but the models do not have knowledge about their execution semantics. We also experimented with adding the API definitions in the prompt, but it didn't increase the performance significantly A.11. Models handle better for predicting Boolean values such as the output of a comparison statement, and also statements where constant is assigned to a variable. The models may understand the assignment operator "=" and have captured easy patterns like "a=3 indicates `a` has value 3 after executing this statement". We did not observe significant advantage of reasoning models over non-reasoning models for this task.

## 3.3 RESULTS FOR RQ3: CODE PROPERTIES WITHIN A FUNCTION

In Fig. 5a, we report results on predicting the number of loop iterations, values in and after the loop, given the input of the function. We observe that models find it difficult to predict values after executing the loop. Loop iterations are the easiest tasks among the three. Our intuition is that

sometimes certain patterns in the code text are linked to the loop iterations. For example, Python code `for i in range(100):.` implies that loop iteration is 100. Somehow, some models know these patterns and constants are linked to the loop iterations. We inspected the predicted loop values and did not find a trend that the models just use any constant numbers in the code text as their answers.

In Fig. 5b and Fig. 5c, we show that given an input, predicting pointer aliasing at a program location and whether a branch can be taken is easier than loop properties. Here, the models only need to predict "yes"/"no". The models predict pointer aliasing better than branch execution. We believe that the models are able to connect code patterns such as "p=q" to the aliasing definition provided in our prompt "when two pointers store the same memory addresses, they are aliasing". Notably, some open-source models perform below 50% on these binary classification tasks—worse than random guessing.

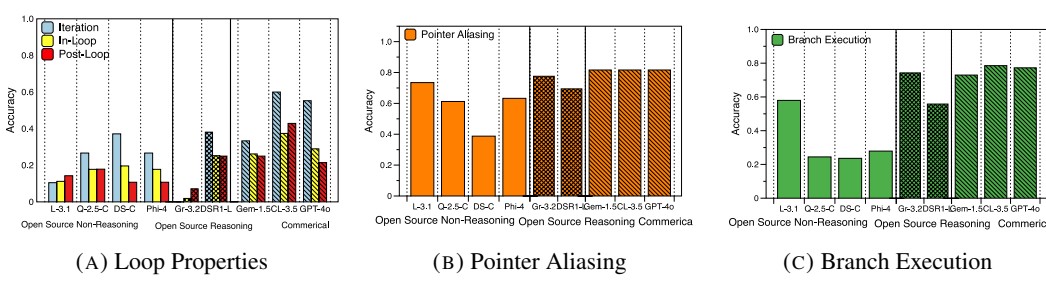

(A) Loop Properties          (B) Pointer Aliasing          (C) Branch Execution

FIGURE 5. **RQ3:** Can models reason about different program properties?

### 3.4 RESULTS FOR RQ4: DIFFERENT PROMPTING TECHNIQUES

In Fig. 6, we show results of different prompting techniques on statement prediction and loop property predictions (relatively difficult tasks in our list). Our results show that in both cases, models benefited from more shots in the prompt. When we prompt models and provide examples more relevant to the query (RAG style); that is, for statement prediction, we provide shots with the same type of statement, and for loops, we provide shots of different loops in the same function, models improved their performance. However, applying a simple COT by "asking models to think step by step" at the beginning of the prompt did not help much for statement prediction, but helped for loop property prediction for some models. Our intuition is that compared to statement prediction, loop reasoning, e.g., predicting values after a loop, may be more complex and can benefit from multi-step reasoning.

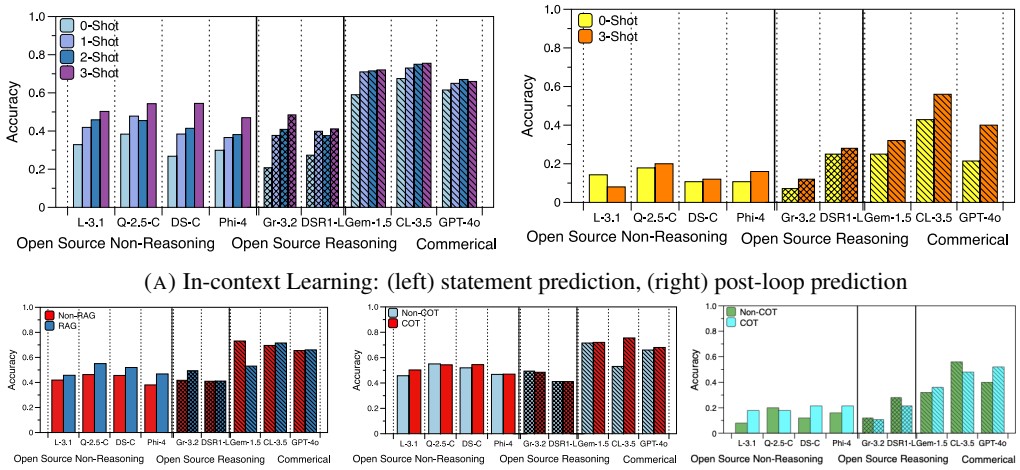

(A) In-context Learning: (left) statement prediction, (right) post-loop prediction

(B) RAG & COT: (left) comparing random shots and shots relevant to the query for statement prediction, (mid) COT for statement prediction, (right) COT for post-loop prediction

FIGURE 6. **RQ4:** Can different prompting strategies help?

## 3.5 RESULTS FOR RQ5: APPROXIMATION OF CODE SEMANTICS

In Fig. 7, we observed that the models reported better performance to predict an approximation of code semantics, for both statement (Fig. 7b) and loop (Fig. 7c) predictions. See also Table 8 in Appendix A.8.3 for comparing with random baselines. Interestingly, when we provide only the definition of "abstract" values (mapping from a range of concrete values to an abstract value) in the prompt , without giving an example showing an "abstract" output for a given input, the models cannot predict abstract values better than concrete values (Fig. 7a). Most models failed to apply definitions directly to the query examples; however, when we provide 3-shots of examples in the prompt, all the models can predict abstract values better than concrete ones.

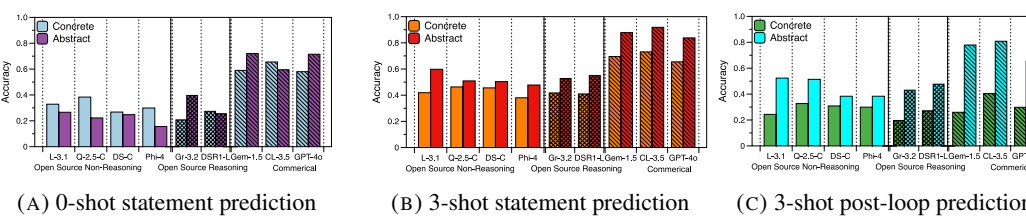

(A) 0-shot statement prediction     (B) 3-shot statement prediction     (C) 3-shot post-loop prediction

FIGURE 7. **RQ5:** Can models reason about an approximation of code semantics? (Python results)

## 3.6 RESULTS FOR RQ6: DIFFERENT PROGRAMMING LANGUAGES

Using input/output prediction as a case study, we investigated models code reasoning capabilities for different programming languages. Fig. 8a shows that Java and Python performed better than C when predicting output given input. Our intuition is that compared to the C code, Java and Python code are more high-level and closer to the natural languages than C; also probably models have seen less C code than Python/Java code in the training data. However, the models reported the lowest accuracy for input prediction of Python code (Fig. 8b).

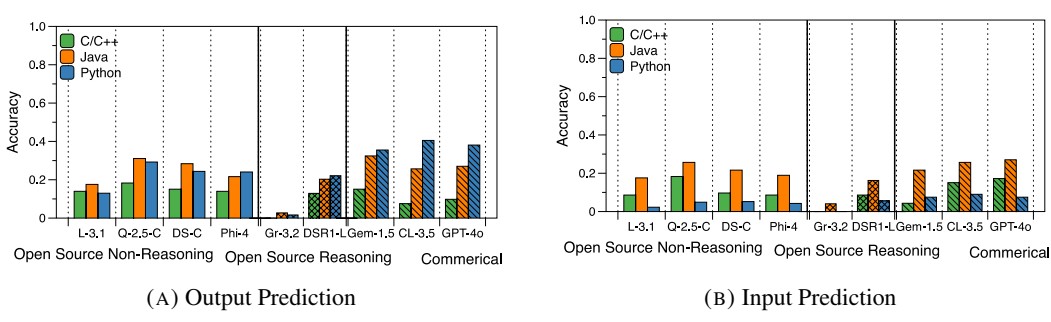

(A) Output Prediction            (B) Input Prediction

FIGURE 8. **RQ6:** Any particular programming languages are easier for the models?

## 4 RELATED WORK

**Code Reasoning Benchmarks.** CruxEval (Gu et al., 2024) assesses LLMs' performance on synthetic Python programs using the task of input-output prediction for a function. CruxEval-X (Xu et al., 2025) extends this work to multilingual settings by translating synthetic Python programs in CruxEval to other languages using LLMs. REval (Chen et al., 2024) evaluated branch prediction tasks using ClassEval (Du et al., 2024) and HumanEval (Chen et al., 2021). CodeMind (Liu et al., 2024) proposed output prediction and code synthesis tasks on existing code benchmarks (Chen et al., 2021; Du et al., 2024; Gu et al., 2024; Ahmad et al., 2023; Austin et al., 2021; Puri et al., 2021). They found that LLM reasoning capabilities deteriorate as program complexity increases (Liu et al., 2024; Zhang et al., 2024b). Our benchmark CodeSense is the first that used real-world Python, C and Java code to evaluate LLMs' reasoning capabilities. While most code reasoning benchmarks reason about function-level execution semantics, we proposed and made ground truth available for a spectrum of fine-grained reasoning tasks regarding program behaviors within a function.

**Other Code Application Benchmarks.** SWE-Bench (Jimenez et al., 2024) used the task of generating patches to resolve a given GitHub issues for real-world Python projects. SWE-PolyBench (Rashid et al., 2025) extends this work to other programming languages. KGym (Mathai et al., 2024) delivered a benchmark consisting of Linux kernel crash data and evaluated LLMs' capabilities of resolving Linux kernel crashes. These benchmarks focus on task-specific performance rather than fine-grained code semantics understanding. There are also benchmarks for code generation (Li et al., 2024; Zhang et al., 2024a; Yu et al., 2024; Chen et al., 2025; Du et al., 2024) and code completion (Izadi et al., 2024; Ding et al., 2023b). However, most of these datasets—such as BigCodeBench (Zhuo et al., 2024) and CodeBenchGen (Xie et al., 2024) are restricted to a single language (primarily Python) and extracted from isolated competitive programming problems.

## 5 CONCLUSIONS

Code semantic reasoning is foundational for solving many software engineering applications. We propose a novel code benchmark and dataset, CodeSense, extracted from 744 Python, C and Java real-world projects, for evaluating LLMs capabilities of code semantic reasoning. We defined a spectrum of fine-grained code reasoning tasks include value predictions at various granularities of the code and program properties prediction for important code constructs like loops, pointers and branches. We developed a framework and tools that can build, test and trace software projects in different programming languages, and can automatically generate ground truth for fine-grained code semantic reasoning tasks. We conducted a comprehensive study on SOTA LLMs. We found that models in general lack the knowledge of code semantics and face challenges for reasoning about even single statements. In limited cases, models can establish the correlation of code semantics description in natural language with some simple frequent code patterns. We hope our dataset and framework can enable further code semantic benchmarks and provide ground truth for future LLMs post-training.

## 6 ACKNOWLEDGEMENT

This work was supported in part by NSF CCF-2313054, NSF CCF-2313055, NSF CNS-2247370, DARPA/NIWC Pacific N66001-21-C-4018. Any opinions, findings, conclusions or recommendations expressed herein are those of the authors and do not necessarily reflect those of the US Government, NSF, or DARPA.

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

## A    APPENDIX

### A.1    COMPUTATION RESOURCES AND INFERENCE TOOLS

All our experiments were conducted using the following computational resources:

- **GPU:** NVIDIA RTX A6000 with 49GB VRAM
- **Memory Utilization:** 0.9GB GPU memory during inference runs
- **Software Stack:** vLLM  (Kwon et al., 2023) for optimised transformer inference
- **Operations:** Inference-only experiments (no fine-tuning performed)

The inference parameters were controlled through the following configuration 4:

TABLE 4. Inference Configuration Parameters

| Parameter | Value | Description |
|---|---|---|
| temperature | 0.8 | Controls randomness: Lower = more deterministic |
| top_p | 0.95 | Nucleus sampling: Only top 95% probability mass |
| max_tokens | 4096 (default) | Base context window size |
|  | 16384 (For reasoning models) | Extended context for specific models |
| tp_size | 1 | No tensor parallelism |
| dtype | float16 | Half-precision floating point |
| stop | [\n»>, \n$, ...] | Generation stopping tokens |

### A.2    LIMITATIONS AND FUTURE WORK

In this work, we consider exact matching between the model's response and ground truth as correct. In future work, we would like to explore other metrics such as pass@k. However, we did explore models' performance of computing abstract value prediction/approximation of code semantics.

For some RQs, we only evaluated the models on subset of tasks. For example, when comparing different programming languages in RQ6, we used input/output predictions. In the future, we will extend such evaluations to more tasks. When computing pointer alias in RQ3, we used C languages. Future work can also include object aliasing detection for Python and Java.

Additionally, we would like to expand our framework to support project tracing and task-specific benchmark datasets beyond the scope of three languages. This includes adding other languages with diverse domains like (e.g., functional, system language), to enable a more comprehensive evaluation of LLMs.

In future, it will be also be interesting to explore more advanced prompting techniques and even fine-tune the models to further evaluate them.

### A.3    TRACE COLLECTION

In this section, we will discuss the detailed process to generate the C, Python and Java real-world traces.

#### A.3.1    REAL WORLD PROJECT COLLECTION

1. **C Project Collection**: For our research, we wanted to generate trace dataset on real-world projects. We have selected real-world C projects that were curated in the OSS-FUZZ repository. The primary reason behind selecting these projects is that they represent different domains to ensure diversity in software types. This choice also aligns with our research goal to generate a benchmark real-world C trace dataset.

2. **Python Project Collection**: To collect real-world Python projects, we adopted two approaches. 1) We cloned all 1489 repositories from GitHub that appear in the PyPIBugs dataset, which was released in 2021 (Allamanis et al., 2021). 2) To avoid missing popular projects after 2021, we use GitHub API to search for repositories that are marked as mainly written in Python and get the results according to the descending order of the number of stars. To maximise the probability that we can execute them easily with the pytest module, we only consider projects that seemingly have a testing folder at the top or second level. For better compatibility and the reflection of the recent trend of programming styles, we further filtered out projects that have not been updated in the last four years. Finally, we got 544 projects.

3. **Java Dataset Collection**: For Java, we aimed to gather a diverse set of real-world projects to have a comprehensive trace analysis, which aligns with our trace dataset generation objective. We have used EvoSuite (Fraser & Arcuri, 2011) for test suite generation, and the SF110 dataset has been used as it is recommended by EvoSuite. This choice ensures compatibility and a high testing coverage rate.

### A.3.2 HIGH-LEVEL STEPS OVERVIEW OF GENERATING TRACES

**C Trace Collection**

1. **Building Projects**: Building projects before fuzzing is necessary to ensure that different project dependencies are correctly installed and configured, avoiding runtime errors during the fuzzing process. This helps to create a consistent and effective environment for the next fuzzing process.

2. **Fuzzing**: In the fuzzing phase, we executed the fuzzer on the already-built projects to generate the input data corpora. We configure the fuzzing tools with appropriate settings and parameters for each project. This includes specifying input seed files, maximum time for the fuzzer to run and kill delay to maximise code coverage. Throughout the fuzzing process, detailed logs are maintained to track the execution progress and other relevant information. These logs can aid in debugging, result interpretation, and fuzzing outcomes.

3. **Tracing**: Tracing is the most crucial step to have the execution information of real-world projects. We use a tracing framework with the GNU debugger to log the execution of the projects. With the help of the framework, we log function calls, variable values, and other states during the execution of the projects. We start the tracing by setting an entry point for the program, and during the execution of the tracing, we record the different states of the program at various points by logging them into an XML-formatted file for further analysis. Additionally, we have added a tracer timeout to ensure the maximum running time of the tracer, as well as an extra kill delay to ensure the safe exit of the tracer. This ensures the reliability and robustness of the tracing process if any unexpected events occur.

**Python Trace Collection**

1. **Execution**: We execute the collected projects to get traces in a best-effort approach: 1) We scan the common dependency files to install the dependencies into an independent Python environment for each project. 2) We use pytest to execute the test cases in the projects and collect the outputs. 3) We analyze the outputs to identify missing dependency errors and try to install the missing dependencies several times.

2. **Tracing**: We use the PySnooper tool to trace the projects but made the following modifications to it: 1) We only keep traces corresponding to source code files in the project source directories to exclude traces happened in Python built-in functions or third-party dependencies. 2) We expand the representation of user-defined class objects by showing the name and value pairs of their first-level attributes. 3) We save the types of variables in traces instead of just value representations to provide more information for the execution-aware source code modeling.

**Java Trace Collection**

1. **Tracing**: To generate the tracing framework for Java, we integrated Java Debugger to record the execution details of the projects. We logged method invocation, variable values, and different program states during the execution cycle. For Java, we stored the raw trace in JSON format, which was stored in directories specific to each class within the project directories. This helps manage large amounts of data, consequently making it easier to retrieve, analyze and clean it up for further tasks.

TABLE 5. Trace Collection

|        | Real-world projects | Testing Tools | Tracing Tools |
|--------|---------------------|---------------|---------------|
| Python | PyPIbugs+Github (544) | Pytest | Pysnooper |
| C      | OSS-Fuzz (100)      | Fuzzing | GDB |
| Java   | SF-110 (100)        | Evosuite | JDB |

## A.4 Language Selection Rationale and Extensibility

The three programming languages in our benchmark are important and representative for programming language features and real-world applications: C is a low level programming language and useful for building systems, Java has object-oriented programming features, are widely used for building enterprise and web applications, Python is important for data science and AI applications. Our benchmark is extensible, third-parties (as well as ourselves in future) can add more languages. Our scripts, prompts and methodologies can be adapted for new programming languages to (1) select and download programs of a programming language from GitHub, (2) fuzz for generating test inputs, (3) trace and curate ground truth data (4) provide input and parse output when interacting with models. Instead of GDB (for C), Pysnooper (for Python) and Java Debugger Oracle Corporation (for Java), we will need to plug in debugging tools for new programming languages.

You may ask "we have compilers and code execution tools, why do we need models to predict dynamic information"—- Predicting dynamic values is not only useful for executing programs, but required for many other downstream tasks. For example, in Fig. 1, to generate test input that can exercise a true branch, the models need to know how each operator in statements updates the values. The key difference is that: when using code execution tools, we give one input and ask for the output, but in other downstream tasks, we require models to first understand fine-grained semantics and then find inputs that can satisfy certain constraints. Even for compiling and executing programs, we may benefit from LLMs prediction, as compilation and execution can be time-consuming and hard to be configured, especially for legacy code. This ability of LLMs is particularly important when the user cannot run the code snippet, e.g., missing dependencies or unavailable resources. Predicting input/output values as code reasoning tasks have been established by prior research (Gu et al., 2024), (Chen et al., 2021), (Yan et al., 2024). Our work extended prior research by introducing real-world projects and fine-grained tasks supported only by our tracing framework.

## A.5 Detailed Description of Tasks

In this section, we will provide a detailed description of our tasks. A.5.1 provides a comprehensive description of the statement-based evaluation we performed on LLMs. We sampled five types of statements, i.e, Assignment, Arithmetic, Constant, Boolean, and Function Call, and prompted the models about the value after execution of each type of statement given the variable states before executing that statement. For the block prediction task A.5.2, we sampled statements from the start of the code snippet, given the input of the code, we prompted the model to predict the output at the end of the 1st statement, the 2nd statement, and the 3rd statement. For Branch prediction A.5.3, we promoted the model whether a specific branch will be taken or not of a code snippet, given the input of that code snippet.

In the case of A.5.4, we sampled loop statements from the code snippets. For each loop, we first collected the number of iterations of that loop as ground truth, and queried the model about how

many times the loop would be iterated. We also collected and prompted the model regarding the variable state inside the loop body after the n-th interaction. We have named this "In-Loop" prediction. Additionally, we sampled variables after the execution of the whole loop and queried the model regarding the variable value after the execution of the loop body ("Post-Loop" prediction). For input/output prediction A.5.5, we used the approach similar to (Gu et al., 2024). For output prediction, we give the entire code snippet and the input of the code snippet, and vice versa for input prediction.

### A.5.1 STATEMENT PREDICTION TASK

```python
def xldate_from_date_tuple(date_tuple, datemode):
    year, month, day = date_tuple
    data_list = list(data_tuple)
    if datemode not in (0, 1):
        raise XLDateBadDatemode(datemode)

    if year == 0 and month == 0 and day == 0:
        return 0.00
    c = 100
    if not (1900 <= year <= 9999):
        raise XLDateBadTuple("Invalid_year:%r" % ((year, month, day),))
    if not (1 <= month <= 12):
        raise XLDateBadTuple("Invalid_month:%r" % ((year, month, day),))
    if day < 1 \
    or (day > _days_in_month[month] and not (day == 29 and month == 2
         and _leap(year))):
        raise XLDateBadTuple("Invalid_day:%r" % ((year, month, day),))

    Yp = year + 4716
    M = month
    if M <= 2:
        Yp = Yp - 1
        Mp = M + 9
    else:
        Mp = M - 3
    jdn = ifd(1461 * Yp, 4) + ifd(979 * Mp + 16, 32) + \
        day - 1364 - ifd(ifd(Yp + 184, 100) * 3, 4)
    xldays = jdn - _JDN_delta[datemode]
    if xldays <= 0:
        raise XLDateBadTuple("Invalid(year,month,day):%r" % ((year, month
            , day),))
    if xldays < 61 and datemode == 0:
        raise XLDateAmbiguous("Before1900-03-01:%r" % ((year, month, day)
            ,))
    return float(xldays)

xldate_from_date_tuple(date_tuple=(1907, 7, 3), datemode=0)
```

**Assignment Prediction**

- What will be the value of the final output of the statement `year, month, day = date_tuple` given `{'date_tuple': (1907, 7, 3)}` after executing the statement?

**Arithmetic Prediction**

- What will be the value of the final output of the statement `Yp = year + 4716` given `{'year': 1907}` after executing the statement?

**Constant Prediction**

- What will be the value of the final output of the statement `c = 0` given `{'constant': 0}` after executing the statement?

**Boolean Prediction**

- Will the true branch of the statement `if datemode not in (0, 1):` be executed given `{'datemode': 0}`?

**Function Call Prediction**

- What will be the value of the final output of the statement `data_list = list(data_tuple)` given `{'date_tuple': (1907, 7, 3)}` after executing the statement??

### A.5.2 BLOCK PREDICTION TASK

```
def exchange(a, i, j):
    temp = a[i]
    a[i] = a[j]
    a[j] = temp
exchange(a=[0, 100, 200, 0, 0, 0, 0, 0, 0, 0], i=2, j=1)
```

**1-Block Prediction**

- What will be the value of the final output of the statement `temp = a[i]` after executing the statement given the function input `'a'`: `[0, 100, 200, 0, 0, 0, 0, 0, 0, 0]`, `'i'`: `2`, `'j'`: `1`?

**2-Block Prediction**

- What will be the value of the final output of the statement `a[i] = a[j]` after executing the statement given the function input `'a'`: `[0, 100, 200, 0, 0, 0, 0, 0, 0, 0]`, `'i'`: `2`, `'j'`: `1`?

**3-Block Prediction**

- What will be the value of the final output of the statement `a[j] = temp` after executing the statement given the function input `'a'`: `[0, 100, 200, 0, 0, 0, 0, 0, 0, 0]`, `'i'`: `2`, `'j'`: `1`?

### A.5.3 BRANCH TASK

```
1.def xldate_from_date_tuple(date_tuple, datemode):
2.
3.   year, month, day = date_tuple
4.
5.   if datemode not in (0, 1):
6.       raise XLDateBadDatemode(datemode)
7.
8.   if year == 0 and month == 0 and day == 0:
9.       return 0.00
10.
11.  if not (1900 <= year <= 9999):
12.      raise XLDateBadTuple("Invalid_year:%r" % ((year, month, day),))
13.  if not (1 <= month <= 12):
14.      raise XLDateBadTuple("Invalid_month:%r" % ((year, month, day),)
    )
15.  if day < 1 \
16.  or (day > _days_in_month[month] and not(day == 29 and month == 2
    and _leap(year))):
17.      raise XLDateBadTuple("Invalid_day:%r" % ((year, month, day),))
18.
19.  Yp = year + 4716
20.  M = month
21.  if M <= 2:
22.      Yp = Yp - 1
23.      Mp = M + 9
24.  else:
25.      Mp = M - 3
26.  jdn = ifd(1461 * Yp, 4) + ifd(979 * Mp + 16, 32) + \
27.      day - 1364 - ifd(ifd(Yp + 184, 100) * 3, 4)
28.  xldays = jdn - _JDN_delta[datemode]
29.  if xldays <= 0:
30.      raise XLDateBadTuple("Invalid(year,month,day):%r" % ((year,
    month, day),))
31.  if xldays < 61 and datemode == 0:
32.      raise XLDateAmbiguous("Before1900-03-01:%r" % ((year, month,
    day),))
33.  return float(xldays)
34.
35.xldate_from_date_tuple((1907, 7, 3), 0)
```

**Brach Prediction**

- Is line 12, raise XLDateBadTuple("Invalid year: %r" % ((year, month, day),)) executed when xldate_from_date_tuple((1907, 7, 3), 0) is called?

### A.5.4 LOOP TASK

```
1.  def make_version_tuple(vstr=None):
2.    if vstr is None:
3.        vstr = __version__
4.    if vstr[0] == "v":
5.        vstr = vstr[1:]
6.    components = []
7.    for component in vstr.split("+")[0].split("."):
8.      try:
9.          components.append(int(component))
10.     except ValueError:
11.         break
12.   components = tuple(components)
13.   return components
14.
15. make_version_tuple('v0.1.1')
```

**Iteration Prediction**

- How many times will the loop on line 7 execute when `make_version_tu‐ple('v0.1.1')` is called?

**In-Loop Prediction**

- What is the value of `components` in line 9 after 2nd iteration when `make_‐version_tuple('v0.1.1')` is called?

**Post-Loop Prediction**

- What is the value of `components` in line 12 when `make_version_tu‐ple('v0.1.1')` is called?

### A.5.5 INPUT-OUTPUT TASK

```
def cast_tuple(val, length = None):
    if isinstance(val, list):
        val = tuple(val)

    output = val if isinstance(val, tuple) else ((val,) * default(
        length, 1))

    if exists(length):
        assert len(output) == length

    return output
```

**Output Prediction**

- What will be the output of the code given input `{'val':1, 'length':4}`?

**Input Prediction**

- What will be the input of the code given output `(1, 1, 1, 1)`?

### A.6 CONCRETE TO ABSTRACT MAPPING

For the approximation of code semantics, we prompted the models to reason in abstract values, instead of reasoning about the concrete exact value. Table 6 shows the mapping from concrete value to abstract category, following the prior literature (Ding et al., 2023a). When defining these mappings,

we carefully aligned the value ranges for each abstract category with the overall value distribution observed in our benchmark. We evaluated the abstract mapping results against a random baseline, where mapping rules were selected randomly from all available mapping categories.

TABLE 6. Concrete Value to Quantize Value Mapping

| Type | Condition | Category |
|------|-----------|----------|
| Integer | $0 < v \leq 10$ | Positive Regular |
| | $v > 10$ | Positive Large |
| | $v == 0$ | Zero |
| | $-10 \leq v < 0$ | Negative Regular |
| | $v < -10$ | Negative Large |
| Float | $1.0 < v \leq 10.0$ | Positive Regular |
| | $0.0 < v \leq 1.0$ | Positive Small |
| | $10.0 < v$ | Positive Large |
| | $v == 0.0$ | Zero |
| | $-1.0 \leq v < 0.0$ | Negative Small |
| | $-10.0 \leq v < -1.0$ | Negative Regular |
| | $v < -10.0$ | Negative Large |
| String | `len(s) == 0` | Empty String |
| | `len(s) > 0 and s.isalpha()` | Alphabetic String |
| | `len(s) > 0 and s.isdigit()` | Numeric String |
| | `len(s) > 0 and not (s.isalpha() or s.isdigit())` | Mixed String |
| List | `len(lst) == 0` | Empty List |
| | `len(lst) > 0` | Non-Empty List |
| Tuple | `len(tup) == 0` | Empty Tuple |
| | `len(tup) > 0` | Non-Empty Tuple |
| Dict | `len(dict) == 0` | Empty Dictionary |
| | `len(dict) > 0` | Non-Empty Dictionary |
| Set | `len(set) == 0` | Empty Set |
| | `len(set) > 0` | Non-Empty Set |
| Boolean | `True` | True |
| | `False` | False |
| NoneType | `None` | None |

### A.7 PROMPTING TECHNIQUES

The following is a subset of prompts we used to evaluate the models. The rest prompts are shown in our data package.

#### A.7.1 RQ1 PROMPT

**Generalized Statement Execution Prediction Prompt**

```
Here's some {lang} code.  Each example highlights a single
statement of (assignment, branch, or function calls) and
shows you what the variable values look like just before it
runs.
Your goal?  Figure out what the result will be right after
that statement runs.
Here are {shot} examples to walk you through it:  ----------
----------------
```

**Assignment Prediction Prompt**

```
You're given some {lang} code and one specific assignment
line.
Here are the local variables just before that line runs.  Can
you figure out what the value of the assignment will be af-
terwards?
Code Snippet:  {lang} {code}
Statement:  {statement}
Before Values:  {variables}
Answer using <ans></ans> tags, Do not include any extra in-
formation.
```

**Boolean Prediction Prompt**

```
Here's a branch(if)/Boolean statement in {lang}, and the val-
ues of the variables it uses.
Will the branch run?  Answer 'Yes' or 'No'.
Code:  {lang} {code}
Branch Statement:  {statement}
Condition Variables:  {variables}
Answer using <ans></ans> tags, Do not include any extra in-
formation.
```

**Function Call Prompt**

```
Here's a function or API call in {lang} with some parameters.
Based on the inputs, what will it return?
Code:  {lang} {code}
Call:  {statement}
Parameter Values:  {variables}
Answer using <ans></ans> tags, Do not include any extra in-
formation.
```

### A.7.2 RQ2 PROMPTS

---

**Generalized Block Execution Prediction Prompt**

```
Take a look at the {lang} code blocks.  One statement is
highlighted in each.
You'll also see the input values going into the function.
Based on those, try to figure out what the highlighted line
will do.
Here are {shot} examples that show how it works:  ----------
-----------------
```

---

**Block Prediction Prompt**

```
Here's a full function in {lang} and a line of code inside it
we care about.
Given the function's inputs, what value will that line pro-
duce?
Code:  {lang} {code}
Statement:  {statement}
Inputs:  {inputs}
Answer using <ans></ans> tags
```

---

**Generalized input_output prompt**

```
Here's some {lang} code.  You'll either get the inputs or the
outputs, but not both.
Your task is to fill in the missing part--predict the out-
put if you know the input, or figure out what input must've
produced the output.
Check out these {shot} examples for reference:  ------------
--------------
```

---

**Output Prompt**

```
Here's some {lang} code and the inputs passed into it.
What output do you expect from it?
Code:  {lang} {code}
Inputs:  {input}
Answer using <ans></ans> tags
```

---

**Input Prompt**

```
You know the output of a piece of {lang} code.  Can you fig-
ure out what the input must've been?
Code:  {lang} {code}
Output:  {output}
Answer using <ans></ans> tags
```

### A.7.3 RQ3 PROMPTS

**Generalized Loop Prediction Prompt**

```
Let's explore some loops in {lang}.  You'll get the full loop
structure along with the input values used in the code.
I'll ask you questions about how the loop body or post-loop
values behave with those inputs.
Here's how it works with {shot} example(s):  ----------------
-----------
```

**Iteration Prediction**

```
Take a look at this lang loop with some given inputs.
Question:{question}
Code:
{lang}
{code}
Answer using <ans></ans> tags.
```

**In-Loop Prediction**

```
This is a {lang} loop and what the input to the function
looks like.
I'll ask you something about what happens inside the loop
body.
Code:
{lang}
{code}
Question:
{question}
Answer using <ans></ans> tags
```

**Post-Loop Prediction**

```
This is a {lang} loop and what the input to the function
looks like.
I'll ask you something about what happens after the loop
body.
Code:
{lang}
{code}
Question:
{question}
Answer using <ans></ans> tags
```

**Branch Prediction Prompt**

```
Here's a branch (if) block statement in {lang}.
Will the branch run given the function call?  Answer 'Yes' or
'No'.
Code:  {lang} {code}
Question:  {question}
Answer using <ans></ans> tags, Do not include any extra in-
formation.
```

**Alias Prediction**

```
Here's some {lang} code with two pointer variables:
- Pointer A: '{pointer_1}'
- Pointer B: '{pointer_2}'
Do these pointers reference the same memory address?  Answer
"Yes" or "No".
Code:  {lang} {code}
Function Input:  {input}
Question:  Do '{pointer_1}' and '{pointer_2}' in (line
{line_1}) point to the same memory location?
Put your answer in <ans></ans> tags.
```

### A.7.4   RQ4 PROMPTS

**Assignment CoT**

```
Let's figure out the result of the assignment:  '{statement}'
You've got the current variable values:  {variables}
Think through the right-hand side, then update the left-hand
side with the result.
```

**Boolean CoT**

```
Here's the condition:  '{statement}'
These are the variable values:  {variables}
Evaluate the condition.  Is it true or false?  That tells you
if the branch runs.
```

**Function Call CoT**

```
This is the function call:  '{statement}'
With these parameter values:  {variables}
Figure out what the function does and predict the return
value.
```

**Block Prediction CoT**

```
First, trace the execution flow till the highlighted state-
ment {statement} and {input} of the given input,
Then identify the variables associated with the statement
Next, use the trace execution flow to evaluate the statement
What value does the statement produce?
```

**Output Prediction CoT**

```
We're given inputs:  {input}
Walk through the code step by step.
Watch how the values change until we get the final output.
Check that it matches what the function should return.
```

**Input Prediction CoT**

```
We know the output:  {output}
Work backwards--what input could've led to that?
Figure out what had to happen in the code, and reverse it to
get the input.
```

**Iteration Prediction CoT**

```
Start the loop using the initial values.
Check the condition, run the body, update, and repeat.
Keep going until the loop ends.
```

**Loop in-Value CoT**

```
Look at the variables at the start of this iteration.
Go through each line in the loop body.
What happens to the variables by the end?
```

**Loop Post-Value CoT**

```
See why the loop stopped (condition failed).
Check the final values of all changed variables.
What did the last iteration do before ending?
What would be the variable value after loop termination?
```

---

**Overall Statement Prediction Prompt (1-shot) with CoT steps**

```
Here's some Python code.  Each example highlights a single
statement of (assignment, branch, or function calls) and
shows you what the variable values look like just before it
runs.  Your goal?  Figure out what the result will be right
after that statement runs.

Here are 1 example to walk you through it:

--------------------------- EXAMPLE 1:  --------------------
Here's a function or API call in Python with some parameters.
Based on the inputs/parameter values, what will it return?

Code:
Python
{in-context Code}

Function Call:  {statement with function call}
Parameter Values:  {values}

Let's think step by step:
This is the function call:  {statement}
With these parameter values:  {variables}
Figure out what the function does and predict the return
value.
Therefore the final answer is:<ans> {Ground Truth} </ans>

Now, please solve the following new problem.

You're given some Python code and one specific assignment
line.  Here are the local variables just before that line
runs.  Can you figure out what the value of the assignment
will be afterwards?

Code:
Python
{Query Code}

Statement:  {selected statement}

Before Values:  {values}

Answer using <ans></ans> tags, Do not include any extra in-
formation.
```

---

### A.7.5 RQ5 PROMPTS

---
**Statement Prediction Prompt with Abstract Mapping**

```
You're given some Python code and one specific assignment line.
Here are the local variables just before that line runs.  Can
you figure out what the value of the assignment will be after-
wards?

Code:
Python
{Query Code}

Statement:  {selected statement}

Before Values:  {values}

You have to give your value prediction using the given quantiza-
tion rules:  {rules_list}

Answer using <ans></ans> tags, Do not include any extra informa-
tion.
```
---

## A.8 ADDITIONAL RESULTS

### A.8.1 RQ1

Fig. 9 and Fig. 10 depict each model's capability on individual statement types. Fig. 11 and Fig. 12 show the performance of all the models across five types of statements for languages Python and C.

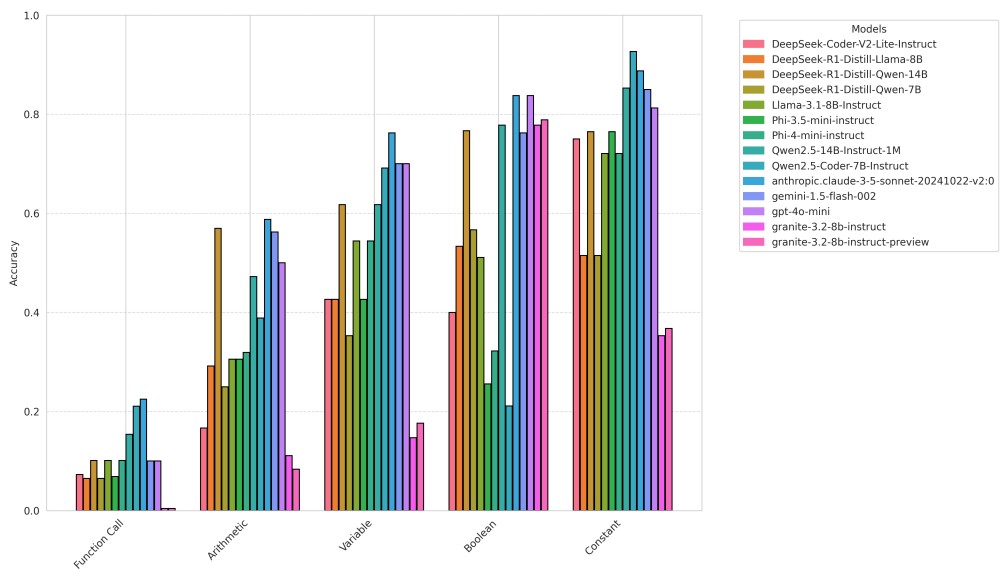

FIGURE 9. **RQ1** Statement type accuracy across Models (Python)

### A.8.2 RQ4

In Fig. 13, we show that for most of the models, adding the number of shots/in-context examples helps the models. Fig. 14 demonstrates that selecting in-context examples in a more controlled way, for example, selecting the same function as in-context examples, helps the models reason better. Finally, Fig. 15 shows whether adding a Chain of Thought (CoT) with the in-context examples can help improve the performance.

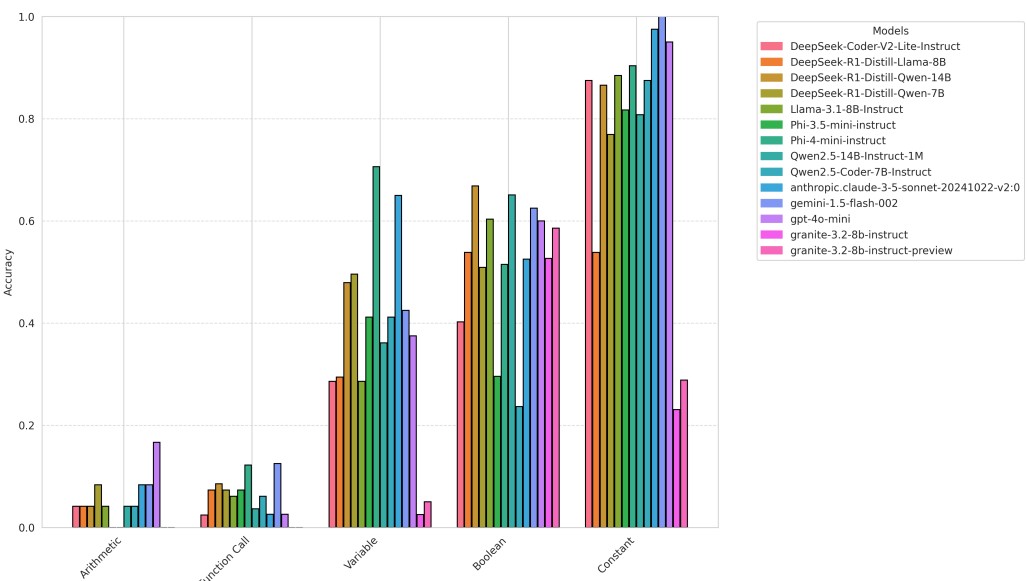

FIGURE 10. **RQ1** Statement type accuracy across Models (C)

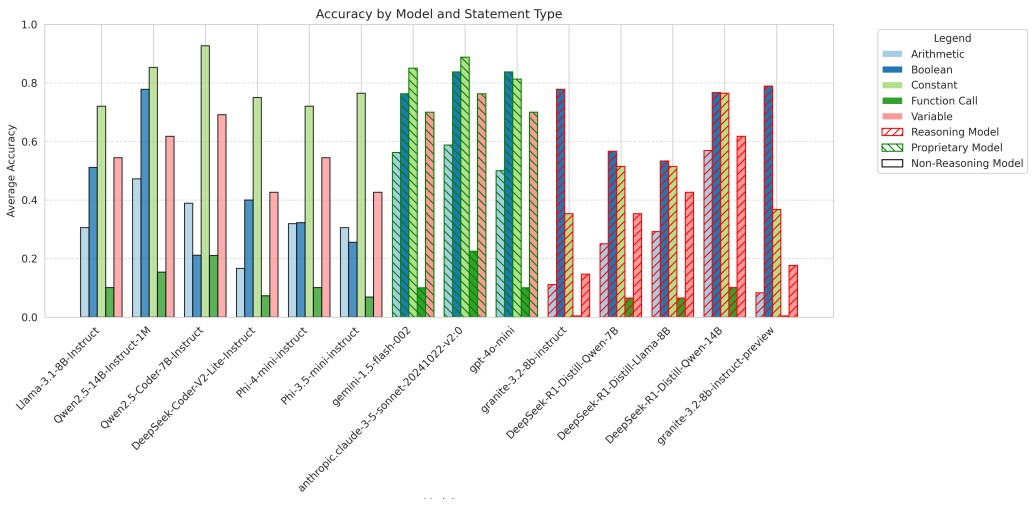

FIGURE 11. **RQ1** Statement type accuracy across Models (Python)

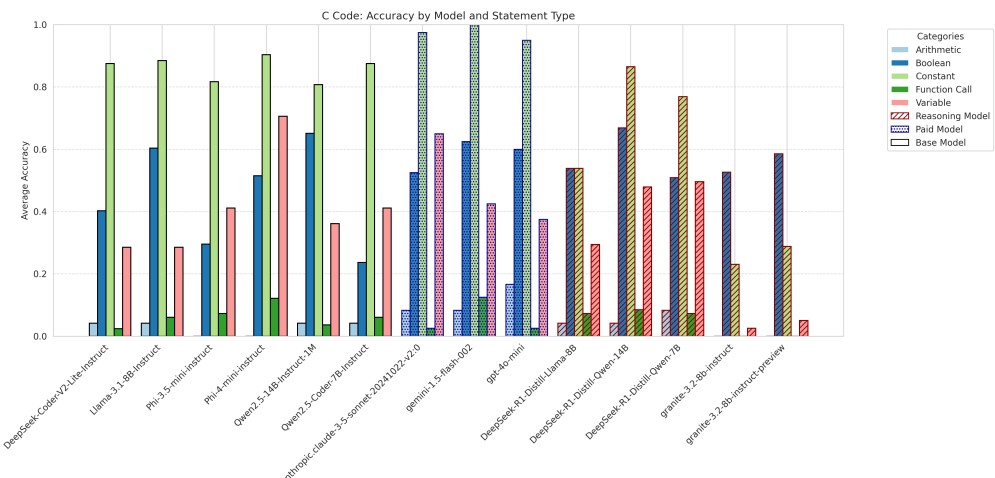

FIGURE 12. **RQ1** Statement type accuracy across Models (C)

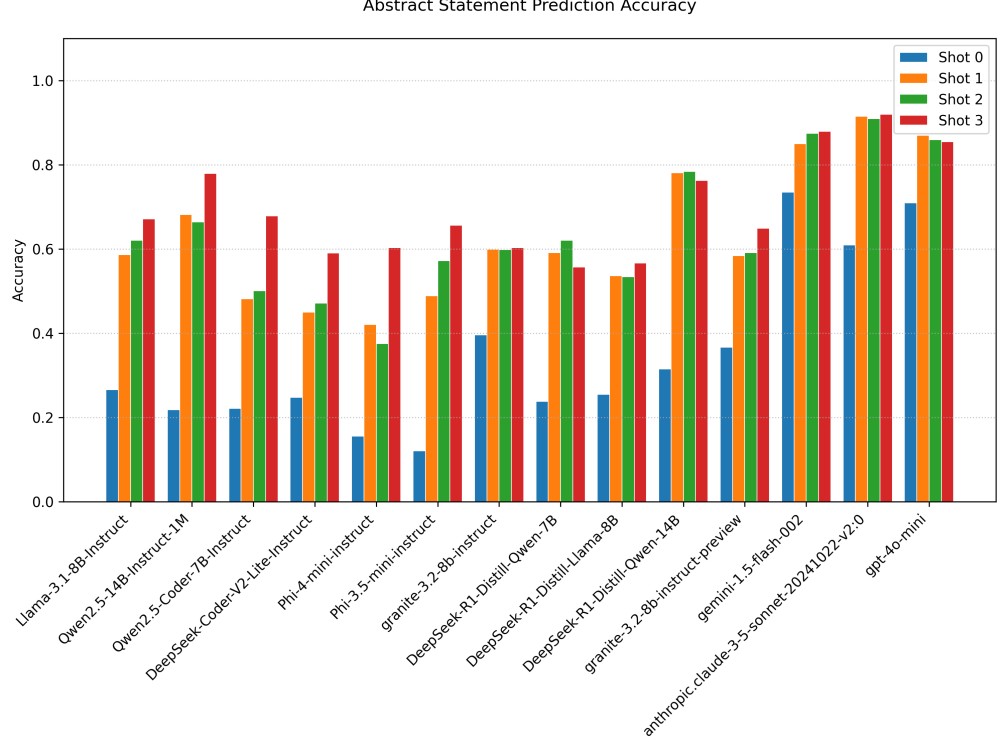

FIGURE 13. **RQ4** Models' Performance with increasing shots from 0 to 3 (Abstract Value Prediction).

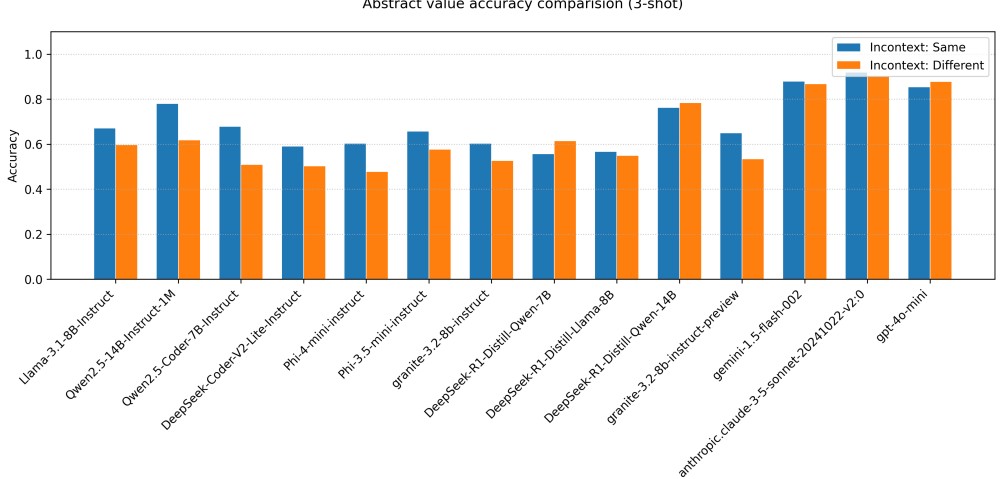

FIGURE 14. **RQ4** Models' Performance with random and same function in-context Examples.

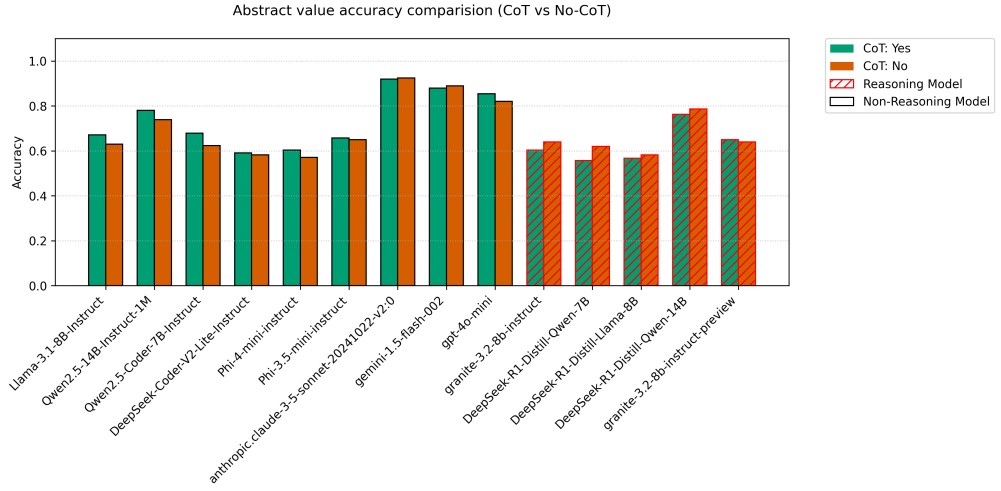

FIGURE 15. **RQ4** Models' Performance CoT vs No-CoT

### A.8.3 RQ5

In Fig. 16, we compare abstract value vs concrete value prediction for post-loop values. Though the models struggle with concrete value prediction, they can improve the performance for predicting the range/approximation of the concrete value.

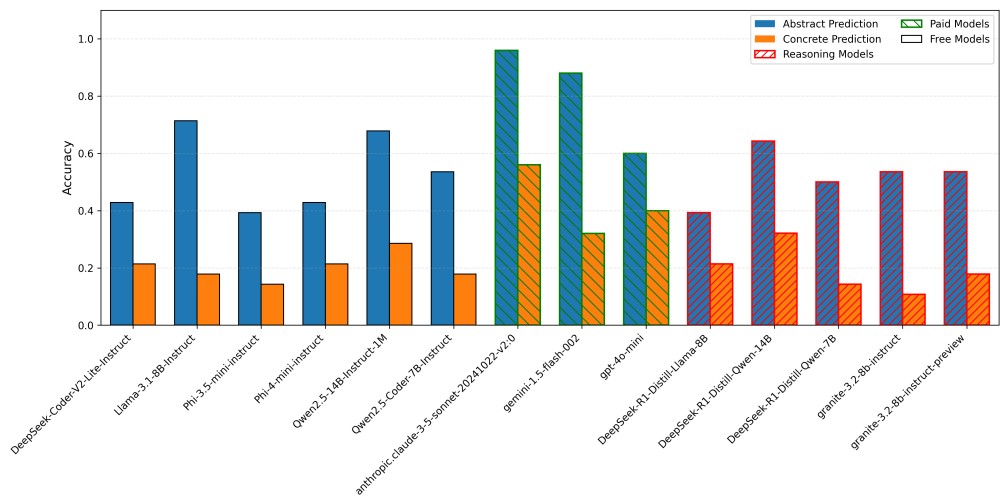

FIGURE 16. **RQ5:** Post-Loop value prediction abstract vs concrete values (3-shots)

### A.9 EXTENDED EVALUATION ON FRONTIER MODELS

To address the concern regarding the performance of the latest frontier models, we added python statement prediction benchmark to include the DeepSeek-V3, DeepSeek-R1 and the recently released Qwen 3 families. Table 7 summarizes these additional results.

### A.10 APPROXIMATION SEMANTICS ON RANDOM BASELINE

The random baseline is calculated by performing a uniform random selection across the possible mapping space for each specific code snippet. The Table 8 shows the models performance on the random baseline against 0-shot and 3-shots.

TABLE 7. Statement Prediction Accuracy for Frontier and Reasoning Models

| Model Name | Statement Prediction Accuracy |
|---|---|
| *Proprietary/Frontier Models* | |
| Anthropic Claude 3.5 Sonnet | 0.65 |
| GPT-4o-mini | 0.60 |
| Gemini 1.5 Flash-002 | 0.60 |
| DeepSeek-V3 | 0.58 |
| *Reasoning Models* | |
| DeepSeek-R1 | 0.53 |
| DeepSeek-R1-Distill-Qwen-14B | 0.42 |
| DeepSeek-R1-Distill-Llama-8B | 0.27 |
| DeepSeek-R1-Distill-Qwen-7B | 0.26 |
| *Qwen 3 Family (New)* | |
| Qwen3-14B | 0.46 |
| Qwen3-8B | 0.36 |

TABLE 8. Model Performance Comparison on Abstract Value Prediction

| Model Name | Random Baseline | 0-shot | 3-shot |
|---|---|---|---|
| DeepSeek-Coder-V2 | 0.084 | 0.247 | 0.504 |
| DeepSeek-R1-Distill-Llama | 0.112 | 0.255 | 0.549 |
| DeepSeek-R1-Distill-Qwen-14B | 0.169 | 0.316 | 0.784 |
| DeepSeek-R1-Distill-Qwen-7B | 0.113 | 0.238 | 0.614 |
| Llama-3.1-8B | 0.104 | 0.266 | 0.597 |
| Phi-3.5-mini | 0.077 | 0.121 | 0.577 |
| Phi-4-mini | 0.088 | 0.156 | 0.478 |
| Qwen2.5-14B | 0.176 | 0.218 | 0.618 |
| Qwen2.5-Coder-7B | 0.099 | 0.222 | 0.509 |
| granite-3.2-8b | 0.152 | 0.396 | 0.528 |
| granite-3.2-8b | 0.147 | 0.367 | 0.535 |

## A.11  API DEFINITION ABLATION STUDY

In Task 2, when predicting the output for output of an API call, we conducted additional experiments to evaluate whether providing API definitions improves model performance. We tested two types of settings: (i) with No API definition and (ii) with API definitions.

We ran our evaluation on the open-source models and evaluated all the 248 function call prediction examples from our dataset. Table 9 shows the results on the best-performing open-source models:

TABLE 9. Accuracy of API prediction with different API definition strategies

| Model | No API definitions | API implementation |
|---|---|---|
| Qwen 2.5-7B | 0.206 | 0.226 |
| Qwen 2.5-14B | 0.182 | 0.194 |
| Phi-4 | 0.125 | 0.089 |
| Llama-3.1-8B | 0.105 | 0.089 |

The results indicate that providing API definitions does not significantly improve performance, and in some cases slightly degrades it. This supports our hypothesis that the fundamental limitation for fine-grained code reasoning is not lack of API knowledge, but rather the models' inability to reason about statement-level and block-level semantics.

## A.12  COMPARISON WITH EXISTING BENCHMARKS

We did a partial evaluation of the output prediction task on our evaluation framework on the three best-performing open-source models on CodeSense, and we present the results in Table 10. Our result shows that the models perform significantly worse in CodeSense than CruxEval.

TABLE 10. Output prediction accuracy comparison: CruxEval vs. CodeSense

| Model | CruxEval | CodeSense | Drop |
|---|---|---|---|
| DeepSeek-R1-Distill-Qwen-14B | 0.75 | 0.37 | 0.38 |
| Qwen2.5-14B | 0.52 | 0.27 | 0.25 |
| Qwen2.5-Coder-7B | 0.50 | 0.30 | 0.20 |

## A.13  VARIANCE ANALYSIS

We have run the statement prediction task, and the results in Table 11 across three runs demonstrate that the variance across multiple runs is minimal, and this doesn't change our core findings.

TABLE 11. Variance analysis across three runs on statement prediction task

| Model | Run 1 | Run 2 | Run 3 | Mean ± Std Dev |
|---|---|---|---|---|
| Qwen2.5-14B | 44.4% | 44.4% | 43.3% | 44.0% ± 0.6% |
| DeepSeek-R1-Distill-Qwen-14B | 42.0% | 41.8% | 43.8% | 42.5% ± 1.0% |
| Qwen2.5-Coder-7B | 38.4% | 38.4% | 38.4% | 38.3% ± 0.0% |
| DeepSeek-R1-Distill-Llama-8B | 26.4% | 25.7% | 27.3% | 26.5% ± 0.8% |

## A.14 DIFFERENT PROMPTING TECHNIQUES FOR STATEMENT PREDICTION

Table 12 shows the difference between two prompting strategies: using in-context examples of the *same* statement type as the query versus examples of a *different* statement type.

TABLE 12. Statement Prediction Performance by Type

| Model | Same Type Statement | Different Type Statement |
|---|---|---|
| Qwen2.5-14B-Instruct-1M | 0.44 | 0.42 |
| DeepSeek-R1-Distill-Qwen-14B | 0.42 | 0.39 |
| Qwen2.5-Coder-7B-Instruct | 0.38 | 0.37 |
| Llama-3.1-8B-Instruct | 0.32 | 0.29 |
| Phi-4-mini-instruct | 0.30 | 0.25 |

## A.15 FUNCTION SIZE ANALYSIS

Table 13 shows how we categorise function difficulties based on lines of code.

TABLE 13. Function Size Categories

| Category | Length (Lines of Code) |
|---|---|
| Small | $length \leq 9$ |
| Medium | $10 < length \leq 19$ |
| Large | $length \geq 20$ |

## A.16 EXAMPLES OF FINE-GRAINED INSIGHTS

Codesense's fine-grained task can uncover reasoning failures on code semantics that are not visible to the coarse-grained benchmarks. For example, consider the simple function from our benchmark

```
def _is_ascii(s):
    if isinstance(s, str):
        for c in s:
            if ord(c) > 255:
                return False
        return True
    return _supports_unicode(s)

_is_ascii(' 123456789#')
```

**Question:** "How many times will the loop on line 3 iterate?"
**Ground Truth:** 11 (The length of the input string `s`, which contains a leading space, digits 1–9, and the # character).

However, models such as `Qwen2.5-Coder-7B` incorrectly respond with **10**. This error reveals that the model fails at a fundamental level: it cannot correctly reason about string iteration, specifically miscounting the characters in a simple string literal.

