# OpenReview forum: "CodeSense: a Real-World Benchmark and Dataset for Code Semantic Reasoning"
_ICLR.cc/2026/Conference — ICLR 2026 Poster_

### Official Review · Reviewer_cZr3 · 2025-11-01

**Soundness:** 3
**Presentation:** 3
**Contribution:** 3
**Rating:** 6
**Confidence:** 4

**Summary:**

The author propose a benchmark CodeSense that makes available a spectrum of fine-grained code reasoning tasks concerned with the software engineering of real-world code. The results show a clear performance gap for the models to handle finegrained reasoning tasks.

**Strengths:**

* The coverage over Python and C are good.

* Sourced from real-world project

* Comprehensive testing over wide range of models.

**Weaknesses:**

* A bit of overselling: Java examples are only 74, which is negligible compared to the total number.

* Figure 3b: What is “smaller” or “larger” function size? Any quantitative number?

* How does author measure accuracy for API call? For something like `time`, it seems impossible to get it right, e.g. time interval for executing code blocks are simply random.

* Can you give example of “abstract values”? For Figure 7, was the few-shot examples with abstract or concrete values?

**Questions:**

* Figure 4: Models are bad at arithmetic so, it seems low accuracy is expected.

* Line 375-376: What if the model is prompted with different type of statement, how would statement prediction perform?

* Could the author include a non-reasoning and reasoning mode for Loop Properties, given the same model (e.g. Qwen3)?

---

> ### Author Response · Authors · 2025-11-25
>
> > A bit of overselling: Java examples are only 74, which is negligible compared to the total number.
>
> This number is the result of filtering from curated total projects (following the processes specified in Sections 2.2 and 2.3).  C and Java real-world projects are much harder to compile and execute, as they require complicated build and execution environments, and therefore, we obtained fewer total projects. We believe this imbalance reflects the nature of real-world settings. We are not willing to add synthetic data for balancing the samples, as we hope the models can be evaluated with real-world scenarios.
>
> We have conducted prompt-based LLM evaluations, where the model handles one function at a time. Their reasoning abilities should not be affected by how many more functions they are going to handle in the pile. For calculating overall accuracy, more data can make the results more confident. Even in some cases, the real-world data are hard to get, we still tried to keep our test examples above 80–100,  as we saw in the published literature \[1].
>
> [1] Saad Ullah, Mingji Han, Saurabh Pujar, Hammond Pearce, Ayse Coskun, and Gianluca Stringhini. Llms cannot reliably identify and reason about security vulnerabilities (yet?): A comprehensive evaluation, framework, and benchmarks, 2024.
>
>
> > Figure 3b: What is the “smaller” or “larger” function size? Any quantitative number?
>
> |                     |              |
> | :-----------------: | :----------: |
> | **Function Length** | **Category** |
> |      Length <=9     |     Small    |
> |  10 < length <= 19  |    Medium    |
> |     Length >= 20    |     Large    |
>
>
> > How does author measure accuracy for API call? For something like time, it seems impossible to get it right, e.g. time interval for executing code blocks are simply random.
>
> We used real-world traces to evaluate the outcome of API calls. We agree that APIs like “time” should be excluded. In our current dataset,  we found out that only 1 sample out of 247 involves a time module API prediction (time.strptime). Thus, our overall results will not be significantly impacted.
>
> > Can you give an example of “abstract values”? For Figure 7, was the few-shot examples with abstract or concrete values?
>
> The few-shot examples were with abstract values. The concrete abstract value mapping table is provided in the appendix table 6.
>
> |  |      |                 |
> | :-------------: | :---------------------------------------------: | :------------------: |
> | **Value Types** |               **Concrete Values**               | **Quantised Values** |
> |     Integer     |                   0 < v <= 10                   |   Positive Regular   |
> |                 |                      v > 10                     |    Positive Large    |
> |                 |                      v == 0                     |         zero         |
> |                 |                   -10 <= v < 0                  |   Negative Regular   |
> |                 |                     v < -10                     |    Negative Large    |
> |      Float      |                 1.0 < v <= 10.0                 |   Positive Regular   |
> |                 |                  0.0 < v <= 1.0                 |    Positive Small    |
> |                 |                     10.0 < v                    |    Positive Large    |
> |                 |                     v == 0.0                    |         zero         |
> |                 |                 -1.0 <= v < 0.0                 |    Negative Small    |
> |                 |                 -10.0 <= v < 1.0                |   Negative Regular   |
> |                 |                    v < -10.0                    |    Negative Large    |
> |      String     |                   len(s) == 0                   |     Empty String     |
> |                 |            len(s) > 0 and s.isalpha()           |   Alphabetic String  |
> |                 |            len(s) > 0 and s.isdigit()           |    Numeric String    |
> |                 | len(s) > 0 and not (s.isalpha() or s.isdigit()) |     Mixed String     |
> |       List      |                  len(lst) == 0                  |      Empty List      |
> |                 |                   len(lst) > 0                  |    Non-Empty List    |
> |      Tuple      |                  len(tup) == 0                  |      Empty tuple     |
> |                 |                   len(tup) > 0                  |    Non-Empty tuple   |
> |       Dict      |                  len(dict) == 0                 |   Empty dictionary   |
> |                 |                  len(dict) > 0                  | Non-Empty dictionary |
> |       Set       |                  len(Set) == 0                  |       Empty set      |
> |                 |                   len(Set) > 0                  |     Non-Empty set    |
> |      Boolen     |                       True                      |         True         |
> |                 |                      False                      |         False        |

---

> > ### Author Response · Authors · 2025-11-25
> >
> > > Figure 4: Models are bad at arithmetic so, it seems low accuracy is expected.
> >
> > There are different types of statements in a program, besides arithmetic. We found that models also cannot predict the boolean value of branch statements, the effect of loop and the outcomes of API calls.
> >
> >
> > > Line 375-376: What if the model is prompted with a different type of statement? How would statement prediction perform?
> >
> > |                              |                     |                          |
> > | :--------------------------: | :-----------------: | :----------------------: |
> > |             Model            | Same Type Statement | Different Type Statement |
> > |    Qwen2.5-14B-Instruct-1M   |         0.44        |           0.42           |
> > | DeepSeek-R1-Distill-Qwen-14B |         0.42        |           0.39           |
> > |   Qwen2.5-Coder-7B-Instruct  |         0.38        |           0.37           |
> > |     Llama-3.1-8B-Instruct    |         0.32        |           0.29           |
> > |      Phi-4-mini-instruct     |         0.30        |           0.25           |
> >
> >
> >
> > > Could the author include a non-reasoning and reasoning mode for Loop Properties, given the same model (e.g. Qwen3)?
> >
> > We thank the reviewer for this excellent suggestion. In our current leaderboard, we have already shown results for reasoning vs non-reasoning models on different tasks, and we agree that running Qwen family models with thinking and non-thinking modes can reveal valuable insights. In the final version of the paper, we will add the evaluation results of the Qwen3 family on all our fine-grained tasks, including all loop properties.

---

### Official Review · Reviewer_C7bm · 2025-11-03

**Soundness:** 1
**Presentation:** 1
**Contribution:** 2
**Rating:** 2
**Confidence:** 3

**Summary:**

Obvious template tampering. Recommended to desk reject.

**Strengths:**

N/A

**Weaknesses:**

N/A

**Questions:**

N/A

---

### Official Review · Reviewer_WPpx · 2025-11-04

**Soundness:** 2
**Presentation:** 3
**Contribution:** 2
**Rating:** 2
**Confidence:** 4

**Summary:**

This paper proposes CodeSense, a benchmark for fine-grained code semantic reasoning constructed from real-world GitHub projects in Python, C, and Java. The benchmark introduces reasoning tasks at statement, code-block, and function levels, targeting semantic properties such as variable values, loop iteration counts, pointer aliasing, and branch conditions. The authors develop an execution tracing framework to automatically generate ground truth from real-world tests, and evaluate 14 state-of-the-art LLMs across six research questions. Results show that models struggle with fine-grained semantic reasoning, particularly on C code and arithmetic operations, with limited improvements from prompting techniques.

This paper addresses an important problem but lacks critical validation of the benchmark's value relative to existing work. While Table 1 compares features with CruxEval, CruxEval-X, REval, and CodeMind, no experiments demonstrate whether CodeSense provides unique insights or advantages that existing benchmarks cannot capture.

**Strengths:**

1. Important problem and motivation: Fine-grained semantic reasoning is crucial for real-world SE tasks like test generation, vulnerability detection, and bug repair. The motivation examples in Figure 1 effectively illustrate this.

2. The paper provides clear presentation and informative figures.

3.Releasing the framework, dataset, and leaderboard supports reproducibility and future work.

**Weaknesses:**

1. Missing Direct Comparison with Existing Benchmarks
The paper claims CodeSense provides advantages over existing benchmarks (Table 1) but provides no experimental validation. Table 1 only compares features (Real-world Projects, Fine-grained Reasoning) without demonstrating whether these features translate to better evaluation quality. The 14 models should be evaluated on both CodeSense and existing benchmarks (CruxEval, REval, CodeMind) to show: (1) whether CodeSense reveals insights that other benchmarks miss, (2) whether fine-grained tasks are more diagnostic than coarse-grained I/O prediction, and (3) whether real-world code introduces challenges absent in synthetic benchmarks. Without this comparison, the benchmark's unique contribution is unproven.


2. Unclear Novelty of Research Findings
The six RQs (RQ1-RQ6) investigate important questions, but without baseline comparison, their novelty is unclear. For example, RQ4 finds "chain-of-thought offers limited benefit" and RQ6 shows "models perform better on Java/Python than C". I may have questions like are these findings unique to CodeSense's fine-grained tasks, or would they also be observed on CruxEval? The paper positions findings as new discoveries without demonstrating what existing benchmarks cannot reveal.


3.The paper does not adequately justify why fine-grained semantic reasoning (statement-level, loop counts, pointer aliasing) is more valuable than existing coarse-grained tasks. While Section 1 motivates the need for semantic reasoning, no evidence shows that fine-grained evaluation better predicts performance on downstream SE tasks. For instance, does a model's ability to predict loop iteration counts correlate with its ability to generate test inputs or detect vulnerabilities? Without this validation, it's unclear whether the added complexity of fine-grained tasks provides practical benefits beyond existing benchmarks.

**Questions:**

Do you have any existing comparison data (even partial) showing how the same models perform on CodeSense vs CruxEval or REval?

Can you provide specific examples of insights that CodeSense reveals but existing benchmarks cannot?

---

> ### Author Response · Authors · 2025-11-25
>
> >Missing Direct Comparison with Existing Benchmark
>
> Please see below the differences between our benchmark and the existing benchmarks. CodeMind and our benchmark are the only two benchmarks that consist of real-world code. Compared to CodeMind, our benchmark focused on statement-level fine-grained code semantics and reasoning of program properties such as loop invariants, branch outcomes.
>
> |  |  | |   |   |   |  |   |                                 |
> | -------------- | :---------------------: | :--------------: | :-------------------------------: | :---------------------------------: | :------------------------------------------------: | :-------------------: | :-----------------------------: | :-----------------------------: |
> | **Benchmark**  | **Real-World Projects** | **Multilingual** | **Function-Level I/O Prediction** | **Fine-Grained Semantic Reasoning** | **Execution-Step / Intermediate State Prediction** | **API Understanding** | **Multi-File Context Handling** | **Realistic Project Structure** |
> | **CruxEval**   |            ✗            |         ✗        |                 ✓                 |                  ✗                  |                          ✗                         |           ✗           |                ✗                |                ✗                |
> | **CruxEval-X** |            ✗            |         ✓        |                 ✓                 |                  ✗                  |                          ✗                         |           ✗           |                ✗                |                ✗                |
> | **REval**      |            ✗            |         ✗        |                 ✓                 |                  ✓                  |                          ✓                         |           ✗           |                ✗                |                ✗                |
> | **CodeMind**   |            ✓            |         ✓        |                 ✗                 |                  ✗                  |                          ✗                         |           ✓           |                ✓                |                ✓                |
> | **CoRE[1]**  |            ✗            |         ✓        |                 ✗                 |                  ✓                  |                          ✗                         |           ✗           |                ✗                |                ✗                |
> | **CodeSense**  |            ✓            |         ✓        |                 ✓                 |                  ✓                  |                          ✓                         |           ✓           |                ✓                |                ✓                |
>
> > Whether CodeSense reveals insights that other benchmarks miss
>
> The existing benchmarks are primarily based on synthetic data and rely heavily on coarse-grained input/output prediction, which basically treats the evaluation as a “black-box” text. The benchmarks are not designed to tell why the prediction failed. We designed our benchmark on real-world projects with fine-grained semantics, which can be used to evaluate the LLMs on strict semantics understanding. Our benchmark asks, even if the model is generating correct input/output, does it understand how the code works?
>
> > Whether fine-grained tasks are more diagnostic than coarse-grained I/O prediction
>
> Yes, statement-level tasks are more diagnostic than I/O prediction.
>
> Our other tasks, such as pointer-aliasing, loop properties and branch outcomes probe different aspects of program semantics compared to I/O predictions.
>
> > Whether real-world code introduces challenges absent in synthetic benchmarks
> Yes. For example, in real-world code, we evaluate if the model can understand API/function calls in the context, CruexEval programs do not have. Our benchmark also includes loop property tasks, pointer aliasing in C codes, and branch prediction. These tasks require a much deeper understanding of code semantics than the available coarse-grained synthetic benchmarks.
>
> We believe our work is not merely scaling up from synthetic to real-world benchmark, but it also reveals the fundamental challenges that are absent in the synthetic benchmark. The benchmark, like CruxEval, is based on small Python codes(3-13 lines), where the complexity of real-world settings is completely missing. For example, these examples are completely based on a limited set of built-in functions or logic, ignoring the real-world API/Function call scenarios. In real-world settings, the LLMs need to understand the implicit knowledge of external APIs, and our RQ2 result shows that the LLMs are poor in API/Function call predictions, an insight that is missing in the current synthetic benchmarks.
>
>
> [1] Danning Xie, Mingwei Zheng, Xuwei Liu, Jiannan Wang, Chengpeng Wang, Lin Tan, and Xiangyu Zhang. Core: Benchmarking llms code reasoning capabilities through static analysis tasks, 2025. URL https://arxiv.org/abs/2507.05269.

---

> > ### Author Response · Authors · 2025-11-25
> >
> > >Unclear Novelty of Research Findings
> >
> > Please see Table 1 for the novelty of our benchmarks. We did a partial evaluation of the output prediction task on our evaluation framework on the three best-performing open-source models on CodeSense, and we present the results in Table 1. Our result shows that the models perform significantly worse in CodeSense than CruxEval.
> >
> > > Better predicts performance on downstream SE tasks
> >
> > We thank the reviewer for raising a valid point regarding how the fine-grained semantics can improve the downstream software engineering tasks. The recent study TRACED [1] shows pre-trained models using dynamic, fine-grained values. Their experimental results showed that even using dynamic values collected from small implementations (<100 lines of code),  the models can improve downstream tasks for real-world code. The downstream tasks evaluated in the paper include branch prediction, vulnerability detection and code clone detection. Another study, SemCoder [2], demonstrates that training the model with code semantics improves the downstream tasks like code generation and execution reasoning.
> >
> > We want to point out that an important contribution of this work is the valuable real-world, multi-lingual datasets and tools/frameworks that can be extended for future benchmarks and training datasets. TRACED [1]  demonstrated that such datasets can be useful for training and improving downstream tasks. But our datasets and tools need to be published first so that we can continuously train models on our real-world execution traces for the downstream SE-related tasks.
> > > Do you have any existing comparison data (even partial) showing how the same models perform on CodeSense vs CruxEval or REval?
> >
> > Table 1: CruxEval vs Codesense (Output Prediction)
> >
> > |                              |              |               |
> > | :--------------------------: | :----------: | :-----------: |
> > |          **Models**          | **CruxEval** | **CodeSense** |
> > | DeepSeek-R1-Distill-Qwen-14B |     0.75     |      0.37     |
> > |          Qwen2.5-14B         |     0.52     |      0.27     |
> > |       Qwen2.5-Coder-7B       |     0.50     |      0.30     |
> >
> >
> >
> > > Can you provide specific examples of insights that CodeSense reveals but existing benchmarks cannot?
> >
> > Yes, Codesense’s fine-grained task can uncover reasoning failures on code semantics that are not visible to the coarse-grained benchmarks. For example, consider the simple function from our benchmark:
> >
> > ```python
> > 1.  def _is_ascii(s):
> > 2.      if isinstance(s, str):
> > 3.          for c in s:
> > 4.              if ord(c) > 255:
> > 5.                  return False
> > 6.          return True
> > 7.      return _supports_unicode(s)
> > 8.
> > 9.  _is_ascii(' 123456789#')
> > ```
> >
> > Question: "How many times will the loop on line 3 iterate?"
> >
> > Ground Truth: 11 (The length of the input string s)
> >
> > However, the models, such as Qwen2.5-coder, incorrectly respond to it as 10. This example shows that the model fails at a basic fundamental level, specifically in reasoning about string iteration.
> >
> >
> > [1] Yangruibo Ding, Benjamin Steenhoek, Kexin Pei, Gail Kaiser, Wei Le, and Baishakhi Ray. Traced: Execution-aware pre-training for source code. In Proceedings of the IEEE/ACM 46th International Conference on Software Engineering, ICSE ’24, New York, NY, USA, 2024. Association for Computing Machinery. ISBN 9798400702174. doi: 10.1145/3597503.3608140.
> >
> > [2] Yangruibo Ding, Jinjun Peng, Marcus J. Min, Gail Kaiser, Junfeng Yang, and Baishakhi Ray. Semcoder: Training code language models with comprehensive semantics reasoning, 2024. URL [https://arxiv.org/abs/2406.01006.](https://arxiv.org/abs/2406.01006)

---

### Official Review · Reviewer_UCFL · 2025-11-05

**Soundness:** 3
**Presentation:** 3
**Contribution:** 3
**Rating:** 6
**Confidence:** 4

**Summary:**

This paper proposes CodeSense, a benchmark for code semantic reasoning sourced from real-world code. The authors curate benchmark samples in Python, C, and Java from GitHub repositories and study a variety of research questions  such as block-level semantics, statement-level semantics.

**Strengths:**

- The benchmark is interesting, addressing the weaknesses of previous benchmarks (CruxEval, REval, CodeMind).
- The authors study a diverse array of code reasoning and program analysis tasks. RQ3 and RQ5 are particularly interesting and have not been studied before (to my knowledge). Many other RQ's are also studied from a fresh perspective, and there is some analysis for each one (some extended in the Appendix)
- A wide variety of models are evaluated and studied, and there is a lot of room for improvement among the studied models.

**Weaknesses:**

- The models studied in the paper are generally weaker than those on the frontier line, even taking into account the lag between the review date and the ICLR submission deadline. The paper would be stronger if it drew insights from failure modes of today's frontier models such as GPT-5, Gemini-2.5-Pro as well as open models like DeepSeek-R1, Qwen3. This would differentiate which of the paper's findings still holds true for the strongest models.
- The research questions are interesting and open up the potential for in-depth exploration, but analysis is only done at a surface level. For example, the CoT's could be analyzed to understand where models are reasoning incorrectly. This is especially interesting for the strongest models.
- Another example is that RQ3 could be studied more carefully, analyzing these trends for different programs (similar to the flavor of https://arxiv.org/abs/2402.05980). RQ5 is very interesting and could also benefit from an analysis of how the type and abstraction level of approximations affects accuracy.
- RQ6 should be studied with paired data, to minimize the effect of confounding variables other than language.
- No variance numbers are reported for the results

**Questions:**

- To what extent do you expect training on traces (e.g. SemCoder, Code World Model) to improve the performance of models? Can you do an analysis to see if these models are significantly better than their non-trace-trained counterparts, or a simple experiment to measure this effect?
- In your opinion, what kinds of insights does your benchmark enable that were not apparent from previous benchmarks?
- In RQ6, how do you know the results are due to the language difference rather than the difference of the problems in the language?
- How do the findings in this work relate to previous work? For example, CruxEval-X studies something very close to RQ6, can you compare the findings? I believe some other RQ's have works studying similar questions.

---

> ### Author Response · Authors · 2025-11-25
>
> >The models studied in the paper are generally weaker than those on the frontier line, even taking into account the lag between the review date and the ICLR submission deadline.
>
> Thank you. We added Deepseel-V3 and Deeseel-R1 to our statement prediction benchmark. See below. The result shows that Deepseel-V3 shows almost a similar result as GPT-40 and Gemini 1.5, and Deepseek-R1 is lagging behind with 53% accuracy. We plan to add all the frontier models like GPT 5.1, Gemini 3 and Qwen 3 family in the final version, and we will continue adding new models to our public leaderboard.
>
>
>
> |                              |                                   |
> | :--------------------------: | :-------------------------------: |
> |        **Model Name**        | **Statement Prediction Accuracy** |
> |          Deepseek-V3         |                0.58               |
> |          Deepseek-R1         |                0.53               |
> |          gpt-4o-mini         |                0.60               |
> |     gemini-1.5-flash-002     |                0.60               |
> |  anthropic.claude-3-5-sonnet |                0.65               |
> | DeepSeek-R1-Distill-Qwen-14B |                0.42               |
> | DeepSeek-R1-Distill-Llama-8B |                0.27               |
> |  DeepSeek-R1-Distill-Qwen-7B |                0.26               |
>
>
> > But analysis is only done at a surface level + Another example is that RQ3 could be studied more carefully, analyzing these trends for different programs (similar to the flavor of https\://arxiv.org/abs/2402.05980). RQ5 is very interesting and could also benefit from an analysis of how the type and abstraction level of approximations affect accuracy.
>
> Agreed. However, In the current paper, our contributions are valuable real-world, multi-lingual datasets/benchmarks and tools/frameworks that can be extended for future benchmarks and training datasets. We demonstrated the struggle of the models in fine-grained code semantics, which provided good examples for performing deeper analysis next step.
>
>
> > RQ6 should be studied with paired data to minimise the effect of confounding variables other than language.
>
> We agree that a controlled paired analysis would help to isolate the pure effect of evaluation on programming syntax and semantics. But the main goal of this RQ was different: to evaluate the models in the real-world settings of naturally occurring codes across different languages. We avoided generating a semantic equivalent synthetic code, as we wanted to evaluate how the models would reason about a real-world project's Python, C or Java functions. It’s important as in real-world settings, the projects are inherently different in coding styles and applications. To ensure a fair comparison despite these inherent differences, we controlled for complexity by filtering our dataset to include only functions with primitive data types in their input parameters and output returns.
>
> > No variance numbers are reported for the results
>
> We ran the experiments. Here are the variance numbers on some of the best-performing open source models:
>
>
> |                              |           |           |           |                    |
> | :--------------------------: | :-------: | :-------: | :-------: | :----------------: |
> |           **Model**          | **Run 1** | **Run 2** | **Run 3** | **Mean ± Std Dev** |
> |          Qwen2.5-14B         |   44.4%   |   44.4%   |   43.3%   |    44.0% ± 0.6%    |
> | DeepSeek-R1-Distill-Qwen-14B |   42.02%  |   41.8%   |   43.8%   |    42.5% ± 1.0%    |
> |       Qwen2.5-Coder-7B       |   38.35%  |   38.35%  |   38.35%  |    38.3% ± 0.0%    |
> | DeepSeek-R1-Distill-Llama-8B |   26.42%  |   25.68%  |   27.34%  |    26.5% ± 0.8%    |
>
> We have run the statement prediction task, and the results across three runs demonstrate that the variance across multiple runs is minimal, and this doesn’t change our core findings. We will add the variance report for key models on all tasks in our final paper.

---

> > ### Author Response · Authors · 2025-11-25
> >
> > >To what extent do you expect training on traces (e.g. SemCoder, Code World Model)
> >
> > The recent study already shows that training the models with traces helps with the downstream software engineering tasks. TRACED \[3] pre-trained models using dynamic, fine-grained values. Their experimental results showed that even using dynamic values collected from small implementations (<100 lines of code),  the models are able to improve downstream tasks for real-world code. The downstream tasks evaluated in the paper include branch prediction, vulnerability detection and code clone detection. It would be really interesting to experiment with how the models would perform if trained on real-world traces. However, conducting rigorous experiments to compare traced-tuned models vs non-traced counterparts is out of the scope of this foundational benchmark paper, as training the model is computationally expensive. But we definitely plan to pursue the model training route in the subsequent work.
> >
> > > Insights
> >
> > The existing benchmarks are primarily based on synthetic data and rely heavily on coarse-grained input/output prediction, which basically treats the evaluation as a “black-box” text. The benchmarks are not designed to tell why the prediction failed. We designed our benchmark on real-world projects with fine-grained semantics, which can be used to evaluate the LLMs on strict semantics understanding. Our benchmark asks, even if the model is generating correct input/output, does it understand how the code works?
> >
> > > In RQ6, how do you know the results are due to the language difference rather than the difference in the problems in the language?
> >
> > We agree with your point. We should update our RQ6 to be “Do models handle Java, C or Python real-world programs better?
> >
> > > How do the findings in this work relate to previous work?
> >
> > Please see Table 1 for comparison with other code benchmarks. The biggest difference is that our benchmark is based on real-world code CruxEval-X, and other recent benchmarks [1], [2], are constructed from models and translated to different languages, defined on coarse-grained I/O prediction tasks. We define more variety of tasks, for example, statement prediction, loop properties prediction, branch prediction, and pointer aliasing property prediction. See below the performance comparison
> >
> >    Table 1: CruxEval vs Codesense (Output Prediction)
> >
> > |                              |              |               |
> > | :--------------------------: | :----------: | :-----------: |
> > |          **Models**          | **CruxEval** | **CodeSense** |
> > | DeepSeek-R1-Distill-Qwen-14B |     0.75     |      0.37     |
> > |          Qwen2.5-14B         |     0.52     |      0.27     |
> > |       Qwen2.5-Coder-7B       |     0.50     |      0.30     |
> >
> >
> > [1] Junkai Chen, Zhiyuan Pan, Xing Hu, Zhenhao Li, Ge Li, and Xin Xia. Reasoning runtime behavior of a program with llm: How far are we?, 2024. URL <https://arxiv.org/abs/2403.16437>.
> >
> > [2] Alex Gu, Baptiste Rozière, Hugh Leather, Armando Solar-Lezama, Gabriel Synnaeve, and Sida I. Wang. Cruxeval: A benchmark for code reasoning, understanding and execution, 2024. URL:<https://arxiv.org/abs/2401.03065>.
> >
> > [3] Yangruibo Ding, Benjamin Steenhoek, Kexin Pei, Gail Kaiser, Wei Le, and Baishakhi Ray. Traced: Execution-aware pre-training for source code. In Proceedings of the IEEE/ACM 46th International Conference on Software Engineering, ICSE ’24, New York, NY, USA, 2024. Association for Computing Machinery. ISBN 9798400702174. doi: 10.1145/3597503.3608140.

---

> ### Comment · Reviewer_UCFL · 2025-11-27
>
> Thank you for the response! I decided to maintain my positive score, due to the contribution of the dataset, tasks, and evaluation on many models (now including more frontier ones). To strengthen the paper, I recommend the authors conduct more in-depth analysis in order to reveal more significant insights regarding the semantic reasoning abilities of LLMs on code.

---

### Author Response · Authors · 2025-12-04
**Summary of Rebuttals and Revisions for Paper**

Dear Area Chair and Reviewers,

Thank you for the thoughtful reviews and feedback, which have helped us to improve our paper. Here, we have summarised the key rebuttals and revisions according to the concerns raised by the reviewers.

**Core Contributions:** Our paper introduces Codesense, the first multilingual benchmark curated from _real-world_ projects to evaluate the code reasoning capabilities of the LLMs using fine-grained semantic reasoning tasks (e.g, statement-level prediction, branch prediction, loop property prediction, pointer aliasing prediction, etc). We have also publicly released the dataset, the trace collection tools and the evaluation framework, which allows future work of building benchmarks for code reasoning and fine-tuning models using real-world traces. Some of the earlier work \[1-2] has shown promising results along the direction.



| Benchmark | Real-World Projects | Multilingual | Function-Level I/O Prediction | Fine-Grained Semantic Reasoning | Execution-Step / Intermediate State Prediction | API Understanding | Multi-File Context Handling | Realistic Project Structure|
|-----------|---------------------|--------------|-------------------------------|----------------------------------|------------------------------------------------|--------------------|-----------------------------|-----------------------------|
| CruxEval  | ✗ | ✗ | ✓ | ✗ | ✗ | ✗ | ✗ | ✗ |
| CruxEval-X | ✗ | ✓ | ✓ | ✗ | ✗ | ✗ | ✗ | ✗ |
| REval | ✗ | ✗ | ✓ | ✓ | ✓ | ✗ | ✗ | ✗ |
| CodeMind | ✓ | ✓ | ✗ | ✗ | ✗ | ✓ | ✓ | ✓ |
| CoRE | ✗ | ✓ | ✗ | ✓ | ✗ | ✗ | ✗ | ✗ |
| CodeSense | ✓ | ✓ | ✓ | ✓ | ✓ | ✓ | ✓ | ✓ |


**Summary of Key Rebuttals:**

> Concern: How does CodeSense differ from existing benchmarks, and are there any comparison results between CodeSense and other benchmarks?

Response: We have added a table showing the differences between CodeSense and other recent code reasoning benchmarks (see above). Our benchmark provides fine-grained granularity at the task level, which is currently lacking in existing benchmarks. We also added a new experiment to compare the output prediction results between CodeSense and CruexEval, and our results show in the rebuttal Table 1 that the models perform significantly worse in CodeSense’s real-world examples. Additionally, we have shown a loop example where models struggle to reason about simple string loop iteration.



> Concern: Some analyses (Deeper RQs study, fine-tuning on real-world traces) can strengthen the paper more.

Response: We appreciate the reviewer's suggestion and valuable future research directions. For this current paper, our contributions include a valuable real-world dataset, the trace collection tools, and benchmark tasks design and ground truth curation. Our tools can be extended to generate future training datasets to improve reasoning capabilities in models. The proposed analysis and fine-tuning are important next steps, and we will pursue those in our subsequent work.




> Concern: Evaluations used slightly older models, variance analysis, and reframing the research question

Response: We have added DeepSeek-V3 and DeepSeek-R1 results (See Reviewer UCFL  Rebuttal Table 1). We will continue evaluating the upcoming new frontier models on our benchmark and updating our public leaderboard. As per the suggestion of the reviewers, we have added a table in the rebuttal with multiple runs to report the variance; the minimal variance confirms the stability of our results.

**Updated Paper Draft to Reflect Reviewer Comments**:

- We have addressed the formatting issue raised by the reviewer C7bm.

* We updated Table 1 to show a comparison between existing benchmarks reference to the reviewer WPpx’s Weakness 1.

* We added Table 9 in Appendix A.11 showing comparisons between CruxEval and CodeSense to address the reviewer WPpx’s Question 1, and the reviewer UCFL’s Question 4.

* We added a specific fine-grained example to show our benchmarks insights in Appendix A.15 to reference the reviewer WPpx’s weakness 2.

* We added a variance analysis Table 10 in Appendix A.12 to address reviewer UCFL’s Weakness 5.

* We added a comparison between different prompting styles, Table 11 in Appendix A.13, to address the reviewer CZr3’s Question 2.

* We added a table to show function size categories, Table 12 in Appendix A.14, to address the reviewer CZr3’s Weakness 2.


[1] Yangruibo Ding, Benjamin Steenhoek, Kexin Pei, Gail Kaiser, Wei Le, and Baishakhi Ray. Traced: Execution-aware pre-training for source code. In Proceedings of the IEEE/ACM 46th International Conference on Software Engineering, ICSE ’24, New York, NY, USA, 2024. Association for Computing Machinery. ISBN 9798400702174. doi: 10.1145/3597503.3608140.

[2] Yangruibo Ding, Jinjun Peng, Marcus J. Min, Gail Kaiser, Junfeng Yang, and Baishakhi Ray. Semcoder: Training code language models with comprehensive semantics reasoning, 2024. URL [https://arxiv.org/abs/2406.01006.](https://arxiv.org/abs/2406.01006)

---

### Meta-Review · Area_Chair_CAaW · 2026-01-09

**Summary:**

This work proposes a benchmark for fine-grained code reasoning tasks in real-world scenarios. The reviewers noted (1) the unique value of the proposed benchmark, which addresses the limitations of existing benchmarks and (2) the comprehensive and insightful evaluation results. The main concerns raised by the reviewers are (1) the models evaluated are weaker than the available frontier LLMs; (2) a lack of comparison with existing benchmarks, which leads to the uncertainty of the novelty of this work.

**Reviewer Concerns:**

The rebuttal has addressed various specific questions raised by the reviewers. Regarding the major concerns, the rebuttal provided a clearer comparison between the proposed benchmark and existing ones, which further emphasized its unique contribution. The concern that some frontier LLMs were not included in the evaluation was partly addressed, but several models remain to be added.

**Reviewer Scores:**

Only Reviewer UCFL has participated in the discussion, and they indicated that they would maintain their assessment.

Reviewer WPpx may change the score from 2 to 4, considering the detailed comparison between the proposed benchmark and the existing ones provided during the rebuttal.

Reviewer C7bm did not provide a valid review. They reported potential formatting issues but the AC and PC did not flag the paper for desk rejection.

Reviewer cZr3 is likely to maintain their already positive score, since some of their questions were answered in the rebuttal, while others remain partially unaddressed.

---

### Decision · Program_Chairs · 2026-01-26

Accept (Poster)